EMBO
Molecular Medicine

# Early 5-HT$_6$ receptor blockade prevents symptom onset in a model of adolescent cannabis abuse

Coralie Berthoux[1], Al Mahdy Hamieh[1,†], Angelina Rogliardo[1,†], Emilie L Doucet[1], Camille Coudert[1,2], Fabrice Ango[1], Katarzyna Grychowska[3], Séverine Chaumont-Dubel[1], Pawel Zajdel[3], Rafael Maldonado[4], Joël Bockaert[1], Philippe Marin[1,‡] & Carine Bécamel[1,‡,*] (ID)

## Abstract

Cannabis abuse during adolescence confers an increased risk for developing later in life cognitive deficits reminiscent of those observed in schizophrenia, suggesting common pathological mechanisms that remain poorly characterized. In line with previous findings that revealed a role of 5-HT$_6$ receptor-operated mTOR activation in cognitive deficits of rodent developmental models of schizophrenia, we show that chronic administration of Δ9-tetrahydrocannabinol (THC) to mice during adolescence induces a long-lasting activation of mTOR in prefrontal cortex (PFC), alterations of excitatory/inhibitory balance, intrinsic properties of layer V pyramidal neurons, and long-term depression, as well as cognitive deficits in adulthood. All are prevented by administrating a 5-HT$_6$ receptor antagonist or rapamycin, during adolescence. In contrast, they are still present 2 weeks after the same treatments delivered at the adult stage. Collectively, these findings suggest a role of 5-HT$_6$ receptor-operated mTOR signaling in abnormalities of cortical network wiring elicited by THC at a critical period of PFC maturation and highlight the potential of 5-HT$_6$ receptor antagonists as early therapy to prevent cognitive symptom onset in adolescent cannabis abusers.

**Keywords** 5HT$_6$ receptor; adolescent cannabis abusers; cognitive deficits; mTOR; synaptic transmission

**Subject Categories** Pharmacology & Drug Discovery; Development; Neuroscience

## Introduction

Cannabis is the most commonly used recreational drug worldwide, and the last 30 years have been marked by a dramatic increase in cannabis consumption at an increasingly early age by young people in most developed countries (Hall & Babor, 2000). Epidemiological studies suggest that cannabis abuse during adolescence confers an increased risk for developing later in life psychotic symptoms and neurocognitive alterations reminiscent of those observed in schizophrenia (D'Souza et al, 2009; Evins et al, 2012). Moreover, neuroimaging studies in adolescent cannabis users revealed structural abnormalities and altered neural activity in the prefrontal cortex (PFC) during resting state and several types of cognitive paradigms, suggesting a critical role of this brain region in the core cognitive symptoms associated with cannabis abuse during adolescence (Schweinsburg et al, 2008; Batalla et al, 2013). Likewise, chronic administration of Δ$^9$-tetrahydrocannabinol (THC), the main psychoactive constituent of cannabis, to adolescent rats, induces behavioral alterations and cognitive deficits in adulthood that reflect, at least in part, PFC dysfunction (Schneider & Koch, 2003; O'Shea et al, 2006; Malone et al, 2010; Renard et al, 2013, 2017; Rubino & Parolaro, 2013, 2016; Zamberletti et al, 2014).

The PFC undergoes highly orchestrated maturation processes during adolescence. These include sprouting and pruning of synapses, refinement of circuit connectivity and of various neurotransmitter systems, such as glutamatergic, GABAergic, dopaminergic, and endocannabinoid systems (Spear, 2000; Rubino & Parolaro, 2016). It is conceivable that any interference with these maturational events such as chronic cannabis consumption during adolescence would lead to irreversible alterations of PFC connectivity and functionality that represent a risk factor for neuropsychiatric disorders in adulthood (Arseneault et al, 2004; Stefanis et al, 2004; Renard et al, 2014). Consistent with this hypothesis and underscoring the vulnerability of the adolescent brain to cannabis exposure, chronic administration of THC to adult rats (at similar doses to those injected in adolescent animals) does not reproduce the long-lasting cognitive deficits observed in THC-injected adolescent animals (Spear, 2000; Rubino & Parolaro, 2013).

Among the targets currently under investigation to alleviate cognitive deficits associated with various neuropsychiatric

---

1 IGF, University of Montpellier, CNRS, INSERM, Montpellier, France
2 Department of Adult Psychiatry, Montpellier University Hospital, Montpellier, France
3 Department of Medicinal Chemistry, Jagiellonian University Medical College, Kraków, Poland
4 Neuropharmacology Laboratory, Department of Experimental and Health Sciences, Pompeu Fabra University, Barcelona, Spain
*Corresponding author. Tel: +33 434 35 92 15; Fax: + 33 467 54 24 32; E-mail: carine.becamel@igf.cnrs.fr
†These authors contributed equally to this work as second authors
‡These authors contributed equally to this work as last authors

---

disorders, including schizophrenia, the serotonin 5-HT$_6$ receptor still raises particular interest in view of its high expression level in brain regions involved in mnemonic functions and the pro-cognitive effects of 5-HT$_6$ receptor blockade in a broad range of cognitive paradigms in rodents (Codony *et al*, 2011; Yun & Rhim, 2011). In an effort to identify signaling mechanisms underlying cognition control by the 5-HT$_6$ receptor, we previously demonstrated that receptor-operated activation of mechanistic Target Of Rapamycin (mTOR) in PFC underlies cognitive deficits in two rat developmental models of schizophrenia, namely neonatal phencyclidine administration and rearing in social isolation after weaning (Meffre *et al*, 2012). These results are consistent with the role of non-physiological mTOR activation in cognitive impairment observed in genetic forms of autism spectrum disorders (ASD) and Down's syndrome (Ehninger *et al*, 2008; Ehninger & Silva, 2011; Troca-Marin *et al*, 2012). Likewise, a previous study has shown that acute THC administration to adult mice induces an activation of mTOR that underlies the associated amnesic-like effects (Puighermanal *et al*, 2009).

In addition to its well-described role in cognition, the 5-HT$_6$ receptor has recently emerged as a key regulator of neurodevelopmental processes such as neuronal migration (Dayer *et al*, 2015), neurite growth (Duhr *et al*, 2014), and dendritic protrusion (Rahman *et al*, 2017). Though the role of mTOR, under the control of 5-HT$_6$ receptor, in these neurodevelopmental mechanisms remains to be established, a large body of evidence indicates that the proper structural and functional development of brain circuitry depends on the fine tuning of mTOR signaling that has also a key influence upon synaptic transmission and synaptic plasticity in the mature brain (Swiech *et al*, 2008; Kim *et al*, 2009; Bockaert & Marin, 2015).

In light of these findings, we hypothesized that chronic cannabis abuse during adolescence might induce a persistent non-physiological activation of mTOR in adolescent brain that might lead to abnormalities in PFC maturation and cognitive impairment at the adult stage. The implication of 5-HT$_6$ receptors in these deficits and the underlying alterations of synaptic transmission remain to be established, an issue we have explored in the present study, using mice injected daily with THC between post-natal days (PNDs) 30 and 45 as a model of cannabis abuse during adolescence. We show that this treatment induces a sustained activation of mTOR signaling in PFC, an alteration of both GABAergic and glutamatergic synaptic transmissions and impairment of LTD in the PFC at layer I/V synapses and deficits in novel object recognition, sociability, and social discrimination in adulthood, which are all prevented by early administration (during adolescence) of the 5-HT$_6$ receptor antagonist SB258585 or the mTOR inhibitor rapamycin. Additional experiments were undertaken to characterize THC-induced modifications of intrinsic properties of layer V pyramidal neurons and whether they can be prevented by blocking 5-HT$_6$ receptor-operated mTOR signaling during adolescence.

# Results

## 5-HT$_6$ receptors mediate delayed mTOR activation elicited by THC administration to adolescent mice

We used a preclinical model of cannabis abuse in adolescent mice, consisting of daily administration of THC (5 mg/kg, i.p.) between PNDs 30 and 45 (Fig 1A). This treatment induced in adulthood a marked increase in the phosphorylation level of mTOR at Ser$^{2448}$ and of its substrate 70 kDa ribosomal protein S6 kinase (p70S6K) at Thr$^{389}$ in the PFC (Fig 1B) but not in the hippocampus (Appendix Fig S1), compared to control mice injected with saline solution (vehicle), indicative of a sustained activation of mTOR signaling in PFC. As expected, THC administration did not promote mTOR activation in cannabinoid type 1 (CB$_1$) receptor-deficient

**Figure 1. Administration of THC during adolescence to mice induces mTOR activation in the PFC and cognitive deficits in adulthood that depend on 5-HT$_6$ receptors.**

A  Schema of the experimental paradigm used for drug administration. Mice were injected daily with THC (5 mg/kg) or vehicle (Veh) during adolescence, from PNDs 30 to 45. SB258585 (SB, 2.5 mg/kg) or rapamycin (Rapa, 1.5 mg/kg) was administered concomitantly with THC or vehicle. Biochemical and behavioral experiments were performed from PND 60.

B  Top: representative Western blots assessing mTOR phosphorylation at S$^{2448}$ and p70S6K phosphorylation at T$^{389}$ as indexes of mTOR activity in PFC of adult WT mice are illustrated. Bottom: data represent the ratios of immunoreactive signals of the anti-phospho-mTOR (S$^{2448}$) or anti-phospho-p70S6K (T$^{389}$) antibodies to the immunoreactive signal of the anti-β-actin antibody and are expressed in % of values in vehicle-injected WT mice. They are the means ± SEM of results obtained in five mice per group. *$P < 0.05$; **$P < 0.01$, one-way ANOVA followed by Newman–Keuls test.

C  Top: schemas illustrating the behavioral tasks performed in WT mice. Bottom: the plots represent the discrimination index for the novel object recognition task (vehicle: $N = 11$, THC: $N = 11$, THC + SB: $N = 11$, THC + Rapa: $N = 12$), the sociability index (vehicle: $N = 11$, THC: $N = 11$, THC + SB: $N = 11$, THC + Rapa: $N = 12$), and the discrimination index for the social discrimination task ($N = 8$ for each group), measured in each condition. *$P < 0.05$, **$P < 0.01$, one-way ANOVA followed by Bonferroni test (bars and error bars correspond to the mean ± SEM, the dotted line to a discrimination index (exploration of novel object − exploration of familiar object/total object exploration) equal to zero).

D  Top: representative Western blots assessing mTOR phosphorylation at S$^{2448}$ and p70S6K phosphorylation at T$^{389}$ in the PFC of adult 5-HT$_6$$^{-/-}$ mice are illustrated. Bottom: data represent the ratios of immunoreactive signals of the anti-phospho-mTOR (S$^{2448}$) or anti-phospho-p70S6K (T$^{389}$) antibodies to the immunoreactive signal of the anti-β-actin antibody and are expressed in % of values in vehicle-injected 5-HT$_6$$^{-/-}$ mice. They are the means ± SEM of results obtained in four mice per group. n.s. $P > 0.05$, one-way ANOVA followed by Newman–Keuls test.

E  Top: schemas illustrating the behavioral tasks in 5-HT$_6$$^{-/-}$ mice. Bottom: the plots represent the discrimination index for the novel object recognition test (discrimination index: 0.30 ± 0.05, $N = 10$, and 0.31 ± 0.03, $N = 11$, for THC + vehicle and vehicle + vehicle conditions, respectively, $P > 0.05$), the 3-chamber social preference test (sociability index: 0.49 ± 0.04, $N = 10$, and 0.48 ± 0.07, $N = 11$, for THC + vehicle and vehicle + vehicle conditions, respectively, $P > 0.05$) and the social discrimination test (discrimination index: 0.15 ± 0.06, $N = 8$, and 0.21 ± 0.06, $N = 8$, for THC + vehicle and vehicle + vehicle conditions, respectively, $P > 0.05$), measured in each condition. One-way ANOVA followed by Bonferroni test (bars and error bars correspond to the mean ± SEM, the dotted line corresponds to a discrimination index equal to zero).

Source data are available online for this figure.

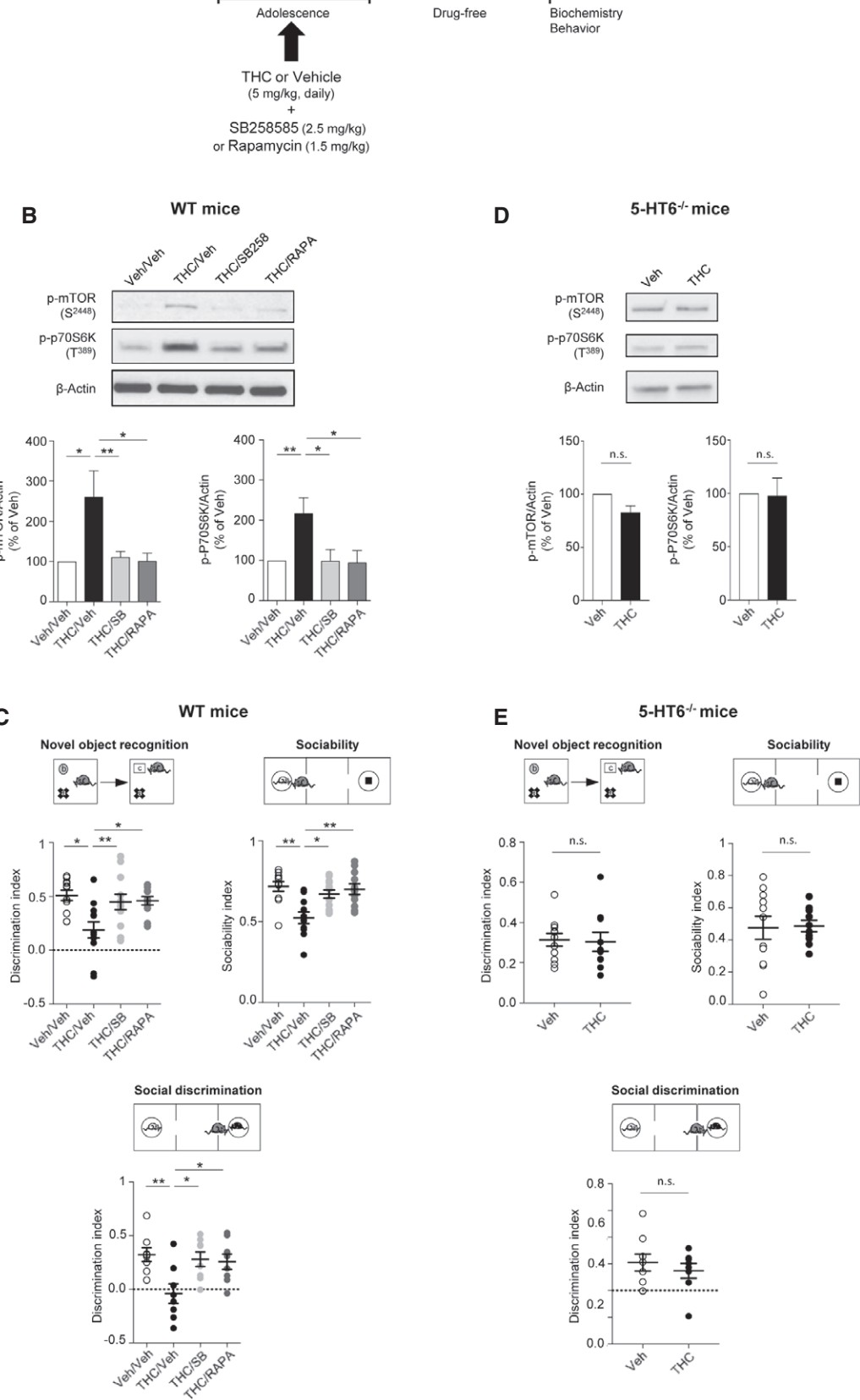

**Figure 1.**

mice ($CB_1^{-/-}$ mice, Fig EV1A), confirming that THC mediates its effects through $CB_1$ receptors. The persistent elevation of phosphorylated mTOR and p70S6K elicited by THC administration during adolescence was prevented by the concomitant injection (between PNDs 30 and 45) of either SB258585 (2.5 mg/kg), a 5-HT$_6$ receptor antagonist (Hirst *et al*, 2000), or rapamycin (1.5 mg/kg), a pharmacological mTOR inhibitor (Fig 1B). Administration of SB258585 or rapamycin during adolescence did not significantly affect the basal phosphorylation state of mTOR and p70S6K in the PFC of vehicle-injected mice (Appendix Fig S2A). Further supporting the role of 5-HT$_6$ receptors in the THC-mediated effects, administration of THC during adolescence did not induce mTOR activation in the PFC of 5-HT$_6$ receptor-deficient mice (5-HT$_6^{-/-}$ mice, Fig 1D).

5HT$_6$ receptors are known to exhibit a high level of constitutive activity both *in vitro* and *in vivo* (Kohen *et al*, 2001; Purohit *et al*, 2003; Duhr *et al*, 2014; Deraredj Nadim *et al*, 2016) that is inhibited by SB258585 (Duhr *et al*, 2014), which thus behaves as an inverse agonist. Administration of the recently characterized 5HT$_6$ receptor neutral antagonist CPPQ ((S)-1-[(3-chlorophenyl)sulfonyl]-4-(pyrrolidine-3-yl-amino)-1H-pyrrolo[3,2-c]quinolone, 2.5 mg/kg) (Deraredj Nadim *et al*, 2016; Grychowska *et al*, 2019) to mice injected with THC during adolescence prevented the persistent elevation of mTOR and p70S6K phosphorylation observed in adulthood (Fig EV1B), suggesting that mTOR activation is due to the activation of 5-HT$_6$ receptors by endogenously released 5-HT rather than agonist-independent activity.

Immunostaining of $CB_1$ and 5-HT$_6$ receptors showed distinct neuronal localizations of both receptors in the mouse PFC. Consistent with previous findings, $CB_1$ receptors are mainly localized presynaptically on GABAergic neurons (as shown by their co-localization with the presynaptic protein Bassoon, Fig 2A, and the GABAergic marker GAD65, Fig 2B; Eggan & Lewis, 2007; Cathel *et al*, 2014) while negligible receptor amounts were detected on 5-HT terminals (Fig 2D). A fraction of $CB_1$ receptors was also found at the postsynapse (co-localization with PSD-95, Fig 2C), corroborating previous findings (Maroso *et al*, 2016). Conversely, 5-HT$_6$ receptors are mostly localized at the postsynapse (Fig 2E–G). Furthermore, no co-localization of $CB_1$ and 5HT$_6$ receptors was found within PFC neurons (Fig 2H). Collectively, these anatomical observations suggest that the long-lasting activation of mTOR signaling in the PFC of THC-injected mice does not result from a crosstalk between $CB_1$ and 5-HT$_6$ receptor-operated signaling but rather from a cross-correlative action on the neuronal cortical network.

## Early blockade of 5-HT$_6$ receptor-operated mTOR signaling prevents cognitive deficits induced by THC administration during adolescence

As previously observed in the rat (O'Shea *et al*, 2006), mice exposed to THC during adolescence showed later in life deficits in behavioral tasks assessing cognitive functions that depend on PFC among other regions: the novel object recognition test (discrimination index: $0.19 \pm 0.07$, $N = 12$, and $0.51 \pm 0.05$, $N = 11$, for THC + vehicle and for vehicle + vehicle conditions, respectively, $P < 0.001$, Fig 1C, left panel), the 3-chamber social preference test (sociability index: $0.71 \pm 0.03$, $N = 11$, and $0.52 \pm 0.03$, $N = 11$, for THC + vehicle and for vehicle + vehicle conditions, respectively, $P < 0.001$, Fig 1C, right panel) and the social discrimination test (discrimination index: $-0.04 \pm 0.09$, $N = 8$, and $0.32 \pm 0.06$, $N = 8$, for THC + vehicle and for vehicle + vehicle conditions, respectively, $P < 0.01$, Fig 1C, bottom panel). In contrast, administration of THC to mice during adolescence did not alter locomotion nor induced an anxiety phenotype in adulthood (Fig EV2).

Given the deleterious influence of non-physiological mTOR activation upon cognition in various neuropsychiatric conditions (Hoeffer & Klann, 2010; Bockaert & Marin, 2015) and its role in cognitive deficits induced by cannabis intake, we next explored whether blocking 5-HT$_6$ receptor-elicited mTOR elevation in adolescent mice exposed to THC prevents the associated cognitive impairments in adulthood. THC-injected mice treated with SB258585 or rapamycin during adolescence showed a similar performance as vehicle-injected animals in the novel object recognition task (discrimination index: $0.45 \pm 0.07$, $N = 12$, and $0.46 \pm 0.04$, $N = 11$, for THC + SB and THC + Rapa conditions, respectively, $P > 0.05$ vs. vehicle/vehicle mice, Fig 1C, left panel), the social preference test (social index: $0.67 \pm 0.02$, $N = 12$, and $0.70 \pm 0.03$, $N = 11$, for THC + SB and for THC + Rapa conditions, respectively, $P > 0.05$ vs. vehicle mice, Fig 1C, right panel) and the social discrimination test (discrimination index: $0.28 \pm 0.06$, $N = 8$, and $0.25 \pm 0.07$, $N = 8$, for THC + SB and for THC + Rapa conditions, respectively, $P > 0.05$ vs. vehicle mice, Fig 1C, bottom panel). SB258585 or rapamycin administration to vehicle-treated mice did not alter their performance in each of these tests (Appendix Fig S2B). Likewise, THC administration to 5-HT$_6^{-/-}$ mice during adolescence did not alter their performance in these three behavioral tasks (Fig 1E). Collectively, these results demonstrate that the chronic intake of THC during adolescence induces a 5-HT$_6$ receptor-dependent

---

**Figure 2. Localization of 5-HT$_6$ and $CB_1$ receptors in the prefrontal cortex.**

A–E The PFC was stained with $CB_1$ receptor (A1–D1, green) or 5HT$_6$ receptor (E1–H1, green), Bassoon (A2 and E2, red), GAD65 (B2 and F2, red), PSD-95 (C2 and G2, red), or SERT (D2 and H2, red) antibodies. Merge images are also depicted (A3–H3). $CB_1$ receptors were mainly co-localized with the presynaptic marker Bassoon (white arrows, A3), mostly within GABAergic boutons (white arrows, B3) and to a lesser extend with PSD-95 (white arrows, C3) and SERT (D3), while 5-HT$_6$ receptors were mainly co-localized with PSD-95 (white arrows, G3). No co-localization of $CB_1$ and 5-HT$_6$ receptors was detected (white arrows indicate the 5-HT$_6$ staining, H3). Scale bar: 10 μm. The scatter plot represents the co-localization analysis for $CB_1$ and 5-HT$_6$ receptor immunostainings. *Graph on the top*. Mander's split coefficient was used to identify the fraction of $CB_1$ receptors that co-localizes with Bassoon (Mander's coefficient: $0.66 \pm 0.02$, $n = 4$), GAD65 (Mander's coefficient: $0.56 \pm 0.01$, $n = 4$), PSD95 (Mander's coefficient: $0.15 \pm 0.01$, $n = 4$), or SERT (Mander's coefficient: $0.082 \pm 0.004$, $n = 4$). *Graph on the bottom*. Mander's split coefficient was used to identify the fraction of 5-HT$_6$ receptors that co-localizes with Bassoon (Mander's coefficient: $0.09 \pm 0.01$, $n = 4$), GAD65 (Mander's coefficient: $0.059 \pm 0.004$, $n = 4$), PSD95 (Mander's coefficient: $0.58 \pm 0.01$, $n = 4$), or CB1 receptors (Mander's coefficient: $0.0020 \pm 0.0004$, $n = 4$). Bars and error bars correspond to the mean $\pm$ SEM.

Source data are available online for this figure.

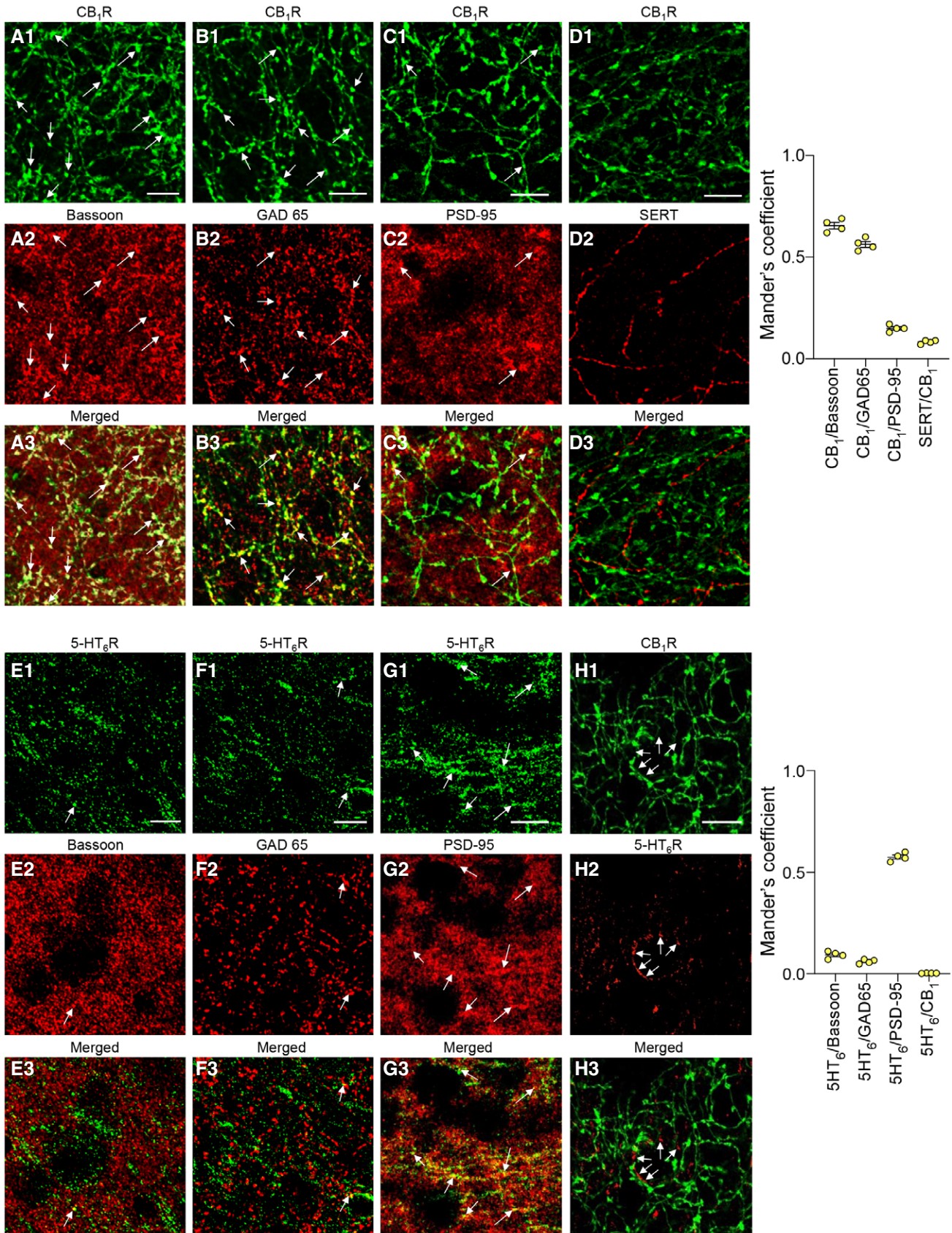

**Figure 2.**

long-lasting activation of mTOR signaling in the PFC that causes cognitive deficits in adulthood.

## Effects of THC administration and of blocking the 5-HT$_6$/mTOR pathway in adulthood

To explore whether blocking 5-HT$_6$ receptor-mediated mTOR activation during adulthood could also have a delayed beneficial influence upon cognition, mice injected with THC during adolescence were treated with SB258585 (2.5 mg/kg) or rapamycin (1.5 mg/kg) during 2 weeks *at the adult stage* (daily injections from PND 60 to 75). Biochemical analysis and behavioral studies were performed 2 weeks after the last injection of the 5-HT$_6$ receptor antagonist or rapamycin (PND 90, Fig 3A). A significant increase in phosphorylated mTOR and p70S6K was observed at PND 90 in THC-injected mice, compared with vehicle-injected mice, and this mTOR overactivation was not affected by SB258585 or rapamycin administration at the adult stage (Fig 3B). Moreover, performances were similar in the THC-injected mice treated or not with SB258585 or rapamycin in adulthood in the novel object recognition task (Fig 3C). These results demonstrate that blocking the 5-HT$_6$/mTOR signaling pathway at the adult stage in mice injected with THC during adolescence does not abolish the long-lasting activation of mTOR and, consequently, does not induce persistent cognitive improvements.

We also administered THC from PNDs 60 to 75 (Fig EV3A) to determine whether its long-term deleterious effects upon mTOR signaling and cognition are really restricted to the juvenile period. Analysis of mTOR signaling 15 days after the last THC injection (PND 90) did not present any change in the phosphorylation state of mTOR and p70S6K in THC-treated mice (Fig EV3B). Correspondingly, THC administration at the adult stage did not induce a persistent deficit in novelty discrimination in the novel object recognition test in the same time frame (Fig EV3C).

## Alterations of GABAergic and glutamatergic synaptic transmissions in THC-injected mice during adolescence are prevented by early blockade of 5-HT$_6$-elicited mTOR activation

To determine whether chronic THC consumption during adolescence affects prefrontal synaptic transmission in adulthood, we analyzed the excitatory and inhibitory synaptic transmissions in acute slices of medial PFC. We performed whole-cell patch-clamp recordings of layer V pyramidal neurons, which integrate excitatory inputs from cortical and sub-cortical areas and measured the amplitude and the frequency of both GABA receptor-mediated miniature inhibitory postsynaptic currents (mIPSCs) and AMPA receptor-mediated miniature excitatory postsynaptic currents (mEPSCs). THC-injected mice showed a robust decrease in mIPSC frequency, whereas their amplitude was not affected (Fig 4B), suggesting a defect in GABA release. Furthermore, in experimental conditions where the inhibitory transmission was initially blocked, mEPSC frequency was significantly increased with no change in their amplitude (Fig 4C). THC-induced alterations of GABAergic and glutamatergic transmissions were abolished by SB258585 or rapamycin administration during adolescence (Fig 4B and C). In mice injected with vehicle during adolescence, neither SB258585 nor rapamycin administration significantly affected mIPSC or mEPSC

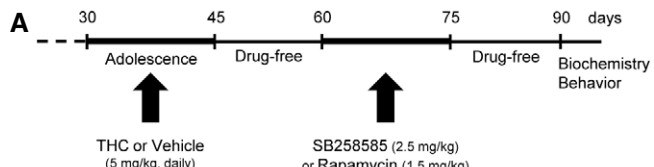

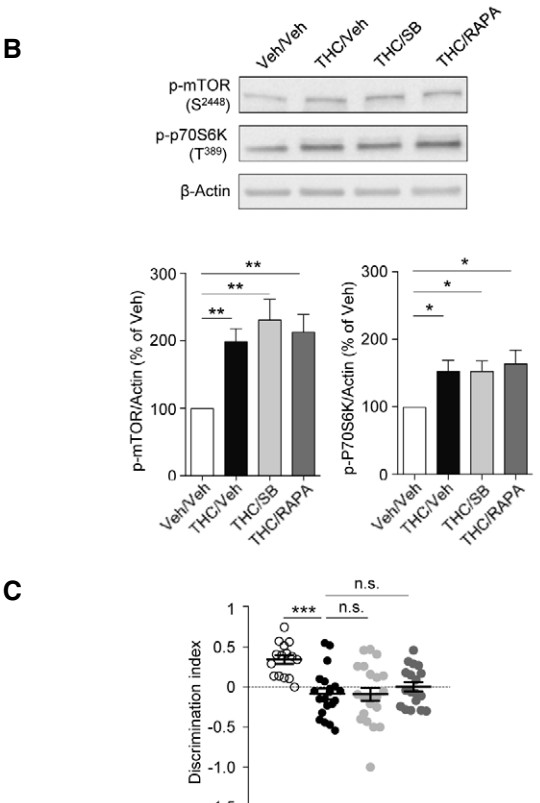

**Figure 3. THC-induced long-lasting mTOR activation and cognitive deficits are not inhibited by the administration of SB258585 or rapamycin in adulthood.**

A   Schema of the experimental paradigm used for drug administration. Mice were injected daily with THC (5 mg/kg) or vehicle (Veh) during adolescence, from PNDs 30 to 45. Vehicle and THC-injected mice were treated daily with either vehicle or SB258585 (SB, 2.5 mg/kg) or rapamycin (Rapa, 1.5 mg/kg) from PNDs 60 to 75. Biochemical and behavioral experiments were performed from PND 90.

B   Top: representative Western blots assessing mTOR activity in PFC are illustrated. Bottom: data represent the ratios of immunoreactive signals of the anti-phospho-mTOR (S$^{2448}$) or anti-phospho-p70S6K (T$^{389}$) antibodies to the immunoreactive signal of the anti-β-actin antibody and are expressed in % of values in vehicle-injected mice. They are the means ± SEM of results obtained in six mice per group. *$P < 0.05$; **$P < 0.01$, one-way ANOVA followed by Newman–Keuls test.

C   The plots represent the discrimination index measured in each condition. ***$P < 0.001$, one-way ANOVA followed by Bonferroni test. The discrimination index for the novel object recognition task is 0.34 ± 0.06 ($N = 14$) and −0.08 ± 0.07 ($N = 20$), for mice injected with vehicle and THC, respectively, $P < 0.001$ and −0.09 ± 0.08 ($N = 21$) and 0.00 ± 0.06 ($N = 18$), for mice treated with THC + SB and THC + Rapa, respectively, $P > 0.05$ vs. THC-injected mice (error bars correspond to the mean ± SEM, the dotted line to a discrimination index equal to zero).

Source data are available online for this figure.

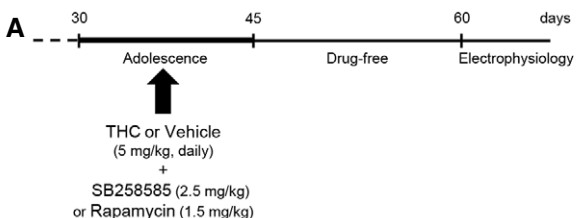

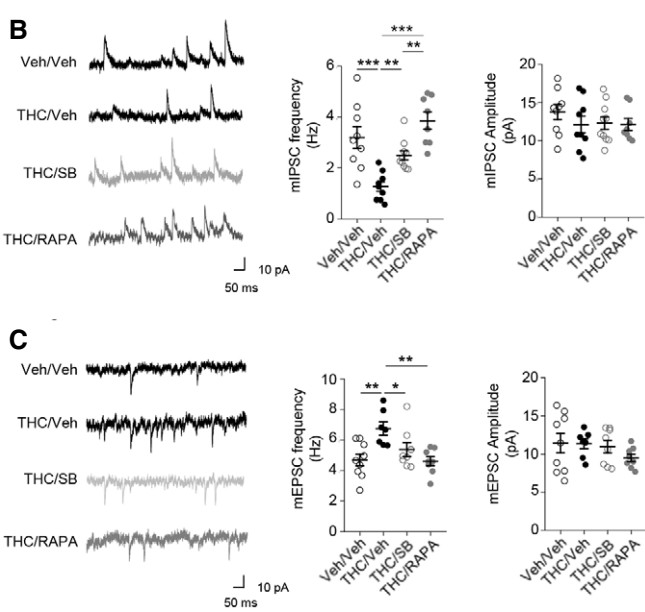

**Figure 4.** THC-induced alterations of prefrontal inhibitory and excitatory synaptic transmissions are prevented by administration of SB258585 or rapamycin during adolescence.

A   Schema of the experimental paradigm used for drug administration. Mice were injected daily with THC (5 mg/kg) or vehicle (Veh) during adolescence, from PNDs 30 to 45. SB258585 (SB, 2.5 mg/kg) or rapamycin (Rapa, 1.5 mg/kg) was administered concomitantly with THC or vehicle. Electrophysiological recordings were performed from PND 60.

B   Left: representative traces of GABA mIPSCs recorded in layer V pyramidal neurons are illustrated. Right: the histograms represent means ± SEM of GABA mIPSC frequency and amplitude measured during the last minute of recording. The mIPSC frequency is: 3.2 ± 0.4 Hz and 1.3 ± 0.2 Hz for vehicle ($n$ = 9 from $N$ = 4) and THC ($n$ = 7 from $N$ = 4) conditions, respectively, $P < 0.001$ and 2.5 ± 0.2 and 4.1 ± 0.4 Hz for THC + SB ($n$ = 8 from $N$ = 4) and THC + Rapa ($n$ = 8 from $N$ = 4) conditions, respectively, $P < 0.01$ and $P < 0.001$ vs. THC-injected mice.

C   Left: representative traces of AMPA mEPSCs recorded in layer V pyramidal neurons are illustrated. Right: the histograms represent means ± SEM of AMPA mEPSC frequency and amplitude measured during the last minute of recording, $n$ = neurons from 3 to 5 mice per group. The mEPSC frequency is 4.7 ± 0.4 Hz for vehicle ($n$ = 9 from $N$ = 7) and 6.8 ± 0.4 Hz for THC ($n$ = 9 from $N$ = 5) conditions, $P < 0.01$ and 5.3 ± 0.5 and 4.6 ± 0.3 Hz for THC + SB ($n$ = 10 from $N$ = 6) and THC + Rapa ($n$ = 9 from $N$ = 5) conditions, respectively, $P < 0.05$ and $P < 0.01$ vs. THC-injected mice. In B and C, *$P < 0.05$; **$P < 0.01$; ***$P < 0.001$, one-way ANOVA followed by Newman–Keuls test.

frequency and amplitude (Appendix Fig S3). These results suggest that cannabis abuse during adolescence induces a sustained alteration of GABAergic and glutamatergic synaptic transmissions in PFC. This leads to a disruption in the excitatory/inhibitory (E/I)

balance that can be prevented by blocking 5-HT$_6$ receptor-operated mTOR signaling.

## THC administration during adolescence induces changes in the intrinsic properties of layer V PFC pyramidal neurons through modulation of HCN1 channels

Spike timing results from dynamic interactions between synaptic activity and intrinsic neuronal excitability. We assessed whether persistent activation of mTOR elicited by THC administration during adolescence might affect the intrinsic electrophysiological properties and the firing rate of layer V pyramidal neurons by monitoring four different parameters: the resting membrane potential (RMP), the action potential (AP) threshold, the rheobase (i.e., minimal current required to induce neuronal firing), and the firing rate (i.e., number of APs for a 150 pA injected current during 250 ms). THC administration during adolescence significantly increased the resting membrane potential (Fig 5A) and lowered the AP threshold (Fig 5B) and the rheobase (Fig 5C), without affecting the firing rate, in adulthood (Fig 5D). Moreover, the concomitant administration of SB258585 or rapamycin prevented these changes (Fig 5A–C) but had no effect on the intrinsic neuronal properties of layer V PFC pyramidal neurons from vehicle-injected mice (Appendix Fig S4). Mimicking the effects of SB258585, concomitant administration of the neutral 5-HT$_6$ receptor antagonist, CPPQ, to THC-injected mice during adolescence also prevented the observed alterations of the resting membrane potential, the rheobase, and the AP threshold (Fig EV4A–C) of PFC pyramidal neurons, without affecting their firing rate (Fig EV4D). Collectively, these results indicate that THC intake during adolescence induces long-lasting changes in the intrinsic neuronal properties of prefrontal layer V pyramidal neurons that are prevented by the early blockade of the 5-HT$_6$/mTOR pathway.

Hyperpolarization-activated cyclic nucleotide-gated channel 1 (HCN1) is the predominant isoform of HCN channels, a family of voltage-gated ion channels responsible for the hyperpolarization-activated current ($I_h$) that modulates spike firing and synaptic potential integration (He et al, 2014; Shah, 2014) by influencing the resting membrane potential of pyramidal neurons. Notably, HCN1 channels enable intrinsic persistent firing of prefrontal layer V pyramidal neurons and are necessary for PFC-dependent behavioral tasks such as executive function during working memory episodes (Thuault et al, 2013). Furthermore, HCN1 channels have been involved in CB$_1$ receptor-induced deficits in LTP in hippocampal pyramidal neurons located in the superficial portion of the CA1 pyramidal cell layer and in spatial memory formation (Maroso et al, 2016). We thus examined whether THC administration during adolescence likewise modifies the activity of HCN1 channels in PFC neurons and measured the voltage sag as an index of postsynaptic $I_h$ in layer V pyramidal neurons, using whole-cell patch-clamp recordings in a current-clamp configuration. Following injection of incremental negative currents (50-pA increment from −400 to 0 pA), the sag amplitude was increased in layer V pyramidal neurons from THC-injected mice, compared with vehicle-injected mice (Fig 5E). Furthermore, administration of SB258585 or rapamycin during adolescence restored a normal sag potential amplitude in THC-injected mice (Fig 5E).

To confirm that HCN1 channels contribute to modifications of the intrinsic neuronal properties, we bath applied onto the PFC

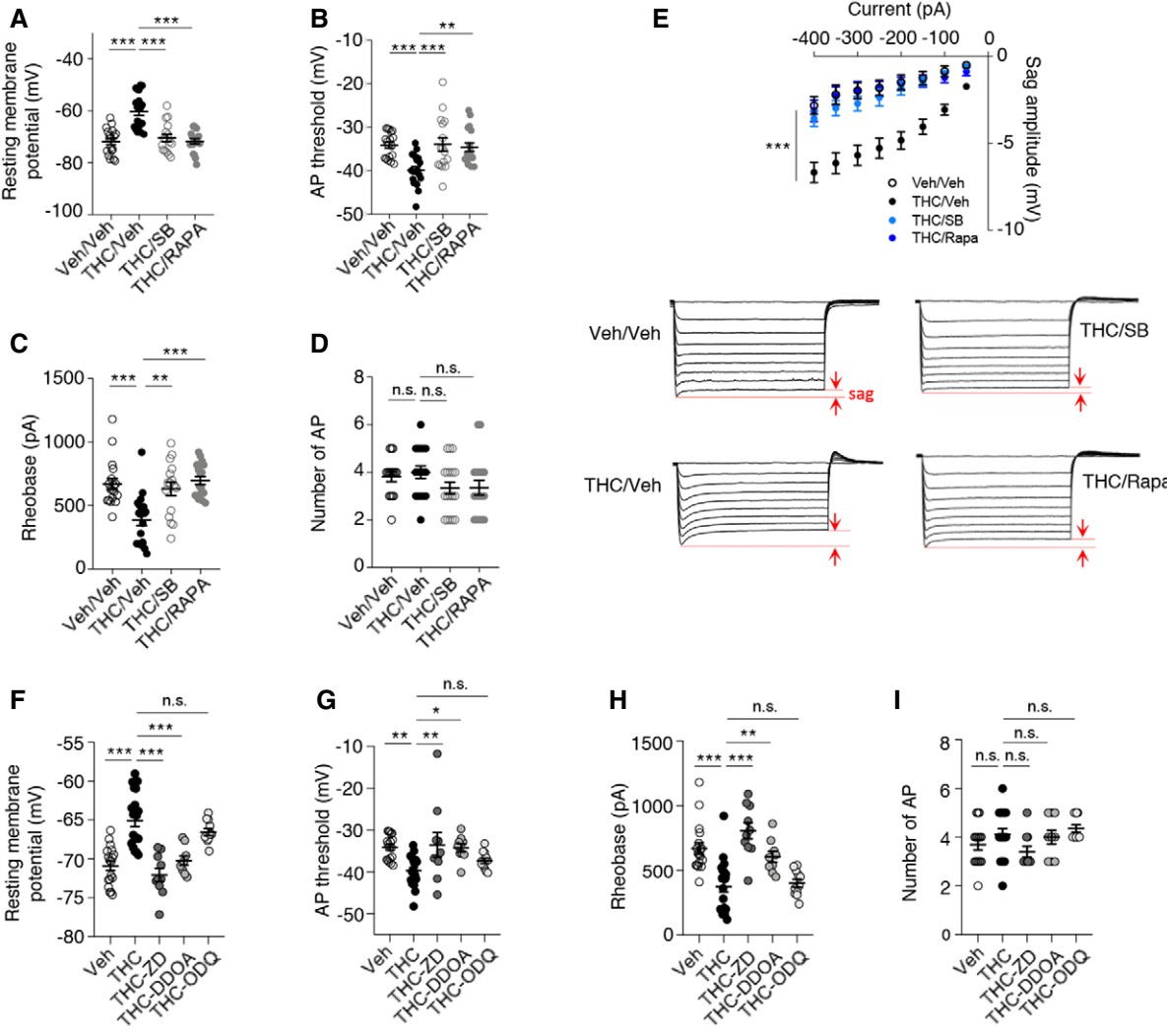

**Figure 5. Administration of THC during adolescence changes intrinsic properties of layer V pyramidal neurons.**

A–D   Mice were injected daily with THC (5 mg/kg) or vehicle (Veh) during adolescence, from PND 30 to 45. SB258585 (SB, 2.5 mg/kg) or rapamycin (Rapa, 1.5 mg/kg) was administered concomitantly with THC (n = 20 from N = 5) or vehicle (n = 18 from N = 6). Electrophysiological recordings were performed from PND 60. (A) The resting membrane potential (RMP) was determined immediately after whole-cell formation: −71.8 ± 1.2 and −60.1 ± 1.5 mV for vehicle and THC conditions, respectively, $P < 0.001$. (B) Action potentials were evoked by a ramp current injection with a 10-pA step for 2 ms, and only the first action potential (AP) was used to estimate the AP threshold: −34.1 ± 0.7 and −39.7 ± 0.8 mV for vehicle- and THC-injected mice, respectively, $P < 0.001$. (C) The rheobase represents the minimal current required to induce neuronal firing, and it was lowered in THC condition: 670 ± 43 and 376 ± 43 pA for vehicle and THC conditions, respectively, $P < 0.001$. (D) The firing rate induced by a 150-pA current injection during 250 ms is expressed as the number of APs. In (A–D), the plots represent means ± SEM of RMPs, AP thresholds, rheobases and firing rates, respectively. ***$P < 0.001$, **$P < 0.01$, one-way ANOVA followed by Bonferroni test. Concomitant administration of SB258585 or rapamycin prevented all the observed changes (RMP: −70.1 ± 1.4 and −72.1 ± 1.0 mV for THC + SB258585 (n = 16 from N = 5) and THC + Rapamycin (n = 18 from N = 5) conditions, respectively, $P < 0.001$ vs. THC-injected mice; AP threshold: −33.0 ± 1.7 and −34.8 ± 1.0 mV for THC + SB258585 and THC + Rapamycin conditions, respectively, $P < 0.001$ vs. THC-injected mice; Rheobase: 624 ± 51 and 686 ± 33 pA for THC + SB258585 and THC + Rapamycin conditions, respectively, $P < 0.001$ vs. THC mice.

E      Top: Voltage sag in response to hyperpolarizing current injection (50-pA increments, from −400 to 0 pA) in PFC pyramidal neurons from mice injected with vehicle (Veh, n = 10 from N = 5), THC (n = 11 from N = 5), THC + SB258585 (THC/SB, n = 7 from N = 4), or THC + Rapa (THC/Rapa, n = 7 from N = 4) during adolescence. ***$P < 0.001$, one-way ANOVA followed by Bonferroni test, vs. THC-injected mice. Errors bars correspond to the mean ± SEM. Voltage sag is indicated by arrowheads.

F–I    Effect of ZD7288 (ZD, 10 μM, n = 10 from N = 3), DDOA (15 μM, n = 9 from N = 4), or ODQ (10 μM, n = 12 from N = 4) on intrinsic properties of PFC pyramidal neurons. THC and vehicle conditions are similar to those on (A–D). (F) resting potential membrane: THC + ZD: −74.14 ± 1.70 mV, $P < 0.001$ vs. THC condition, THC + DDOA: −70.47 ± 1.19 mV, $P < 0.001$ vs. THC condition, THC + ODQ: −63.10 ± 0.81 mV, $P < 0.001$ vs. THC condition, (G) AP threshold: THC + ZD: −33.59 ± 2.96 mV, $P < 0.01$ vs. THC condition, THC + DDOA: −34.28 ± 0.98 mV, $P < 0.001$ vs. THC condition, THC + ODQ: −36.84 ± 0.79 mV, $P < 0.001$ vs. THC condition (H) rheobase: THC + ZD: 806 ± 63 pA, $P < 0.001$ vs. THC condition, THC + DDOA: 607 ± 43 pA, $P < 0.001$ vs. THC condition, THC + ODQ: 422 ± 34 pA, $P < 0.001$ vs. THC condition, and (I) firing rate. Note that vehicle- and THC-injected mice are the same as those used in experiments illustrated in (A–D).
***$P < 0.001$, **$P < 0.01$, *$P < 0.05$, n.s. $P > 0.05$ one-way ANOVA followed by Bonferroni test.

slices an organic HCN1 antagonist ZD7288 at a concentration of 10 μM that minimizes non-specific effects on synaptic transmission (Chevaleyre & Castillo, 2002). Application of ZD7288 restored the resting membrane potential, the AP threshold and the rheobase of layer V pyramidal neurons from THC-injected mice to values close to those observed in vehicle-treated animals (Fig 5F–I), reminiscent of the effects induced by administration of SB258585 or rapamycin during adolescence, whereas it did not modify the intrinsic properties nor the firing pattern of neurons from vehicle-treated mice (Appendix Fig S5). As previously shown (Thuault et al, 2013), the pharmacological inhibition of HCN1 did not significantly alter the firing rate in layer V pyramidal neurons from vehicle-injected animals (Fig 4I and Appendix Fig S5).

HCN1 are gated by both cAMP and cGMP (He et al, 2014). Any increase in cAMP or cGMP would thus cause a depolarizing shift in the activation curve for $I_h$. Consistent with this hypothesis, a recent study has shown that CB$_1$ receptors modulate HCN1 activity and increase postsynaptic $I_h$ through a cGMP-dependent pathway (Maroso et al, 2016). In order to determine whether the alteration of HCN1 activity is mediated by an increase in intracellular cGMP, we perfused 1H-[1,2,4]oxadiazolo[4,3-a]quinoxalin-1-one (ODQ, 10 μM), a selective inhibitor of guanylate cyclase (the enzyme responsible for cGMP production) (Boulton et al, 1995) into the recording pipette. Postsynaptic application of ODQ did not modify the intrinsic neuronal properties of neurons from THC-injected mice (Fig 5F–I). In contrast, perfusion of the selective adenylate cyclase inhibitor 2′,3′-dideoxyadenosine (DDOA, 15 μM; Pelkey et al, 2008) restored the resting membrane potential (Fig 5F), the AP threshold (Fig 5G) and the rheobase (Fig 5H) of layer V pyramidal neurons from THC-injected mice to values similar to those observed in vehicle-treated animals. Application of DDOA or ODQ did not modify the intrinsic properties nor the firing pattern of neurons from vehicle-injected mice (Appendix Fig S5). Collectively, these results suggest a role for cAMP in the HCN1-mediated alterations of neuronal intrinsic properties of PFC pyramidal neurons.

## LTD impairment induced by adolescent THC exposure is prevented by early blockade of 5-HT$_6$ receptors

Previous studies have shown that exposure of rats to THC during adolescence disrupts endocannabinoid-dependent long-term depression (LTD) in the PFC at adulthood (Rubino et al, 2015). Corroborating these findings, we found that electrically induced LTD at PFC layer I/V synapses was impaired in mice treated with THC during adolescence (Fig 6A and B). As expected, the concomitant administration of SB258586 during adolescence restored normal LTD at these synapses (Fig 6A and B), indicating that long-term modifications of synaptic plasticity induced by adolescent exposure to THC depend on 5-HT$_6$ receptor activation during this critical period of PFC maturation.

# Discussion

Previous studies have shown that an acute administration of THC in adult mice induces a transient mTOR activation in the hippocampus that leads to an alteration in protein translation in this brain area and amnesic-like effects in memory tasks depending of hippocampal function (Puighermanal et al, 2009, 2013). In contrast, in the present study, we showed that chronic exposure to THC during adolescence induces a sustained activation of mTOR signaling in the PFC but not in the hippocampus, reminiscent of previous observations in two neurodevelopmental models of schizophrenia, rats treated with phencyclidine at the neonatal stage or reared in social isolation after the weaning (Meffre et al, 2012). These findings are consistent with the hypothesis that common signaling mechanisms might contribute to cognitive symptoms in patients with schizophrenia and cannabis abusers during adolescence. They also suggest that THC intake might differentially influence mTOR activity depending on its mode of administration and the age of delivery: whereas an acute administration at the adult stage induces a transient elevation of mTOR signaling in the hippocampus (Puighermanal et al, 2009,

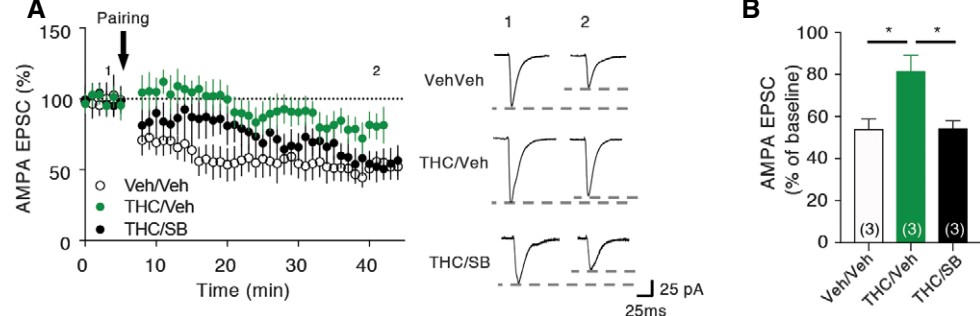

**Figure 6. THC-induced impairment of LTD at PFC layer I/V synapses is prevented by the administration of SB258585 during adolescence.**

A    In left, normalized peak amplitudes of isolated AMPA EPSCs recorded at −60 mV, before, and after pairing protocol, are illustrated. Representative traces of AMPA EPSCs before (1) or after (2) the pairing protocol are also illustrated (right panel) for each experimental condition: vehicle-injected mice (white circles, n = 3 from N = 3), THC-injected mice (green circles, n = 3 from N = 3), and THC-injected mice treated with SB258585 during adolescence (black circles, n = 3 from N = 3). Bars and errors bars correspond to mean ± SEM.

B    The histogram represents the means ± SEM of AMPA EPSCs in % of baseline, measured during the last 5 min of baseline or the last 5 min of recording, for each experimental condition: vehicle-injected mice (white bar, n = 3 from N = 3), THC-injected mice (green bar, n = 3 from N = 3), and THC-injected mice treated with SB258585 during adolescence (black bar, n = 3 from N = 3). n.s. P > 0.05, *P < 0.05, one-way ANOVA followed by Tukey's test.

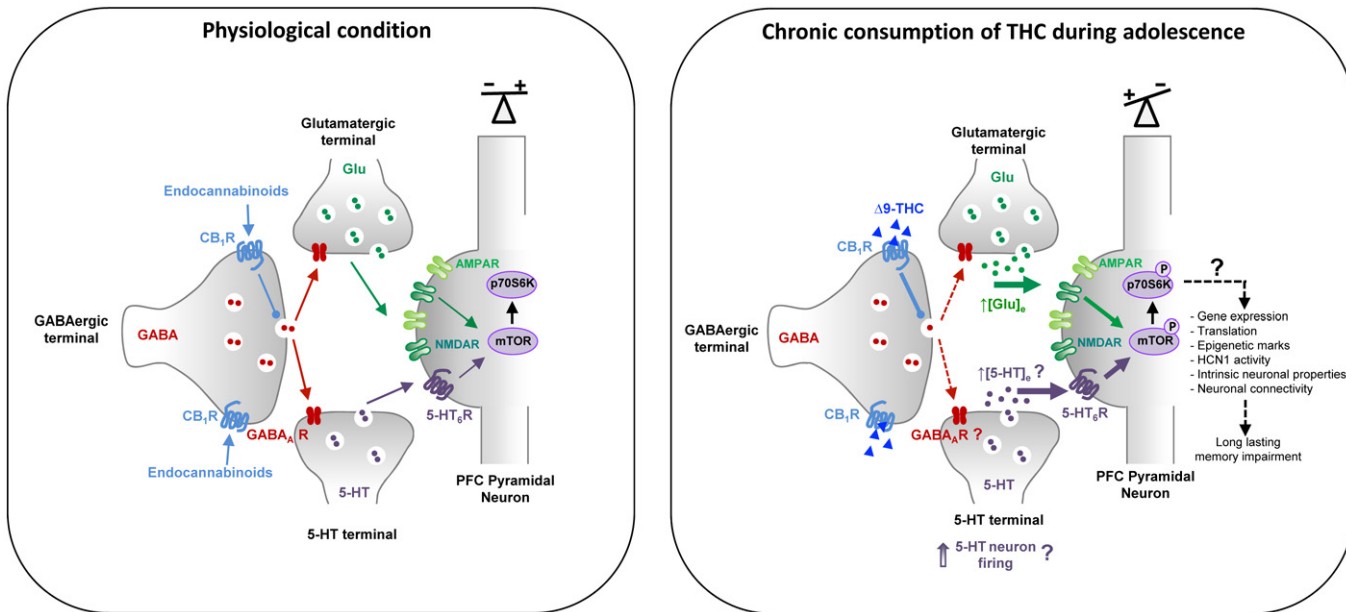

**Figure 7. Model illustrating the possible mechanism induced in the prefrontal cortex by chronic consumption of THC during adolescence that leads to long-lasting cognitive impairment.**

Left panel: physiological condition. Right panel: chronic consumption of THC during adolescence induces an overactivation of prefrontal $CB_1$ receptors that are mainly located on GABAergic terminals. As previously established in adult mice that received an acute THC treatment (Puighermanal *et al*, 2009), this might lead to a decrease in GABA release and, subsequently, to an increase in the glutamatergic tone. Likewise, adolescent THC consumption might also increase serotoninergic tone leading to a sustained activation of $5-HT_6$ receptor located on pyramidal neurons which is prerequisite for the long-lasting stimulation mTOR signaling pathway. This in turn leads to an alteration in intrinsic neuronal properties and cognitive deficits in adulthood. The non-physiological activation of mTOR and resulting imbalance in the excitatory/inhibitory equilibrium during adolescence might also interfere with maturational events occurring in the adolescent PFC leading to a persistent rearrangement of cortical networks affecting the serotonergic system itself and, ultimately, to a persistent mTOR activation. For the clarity of the figure, $CB_1$ receptors were only represented on GABAergic terminals. The presence of GABA-A receptors onto glutamatergic and serotoninergic terminals were previously suggested in Alle and Geiger (2007); Ruiz *et al* (2010); Yamamoto *et al* (2011); and Cerrito *et al* (1998), respectively.

2013), chronic intake during adolescence leads to a long-lasting mTOR activation in the PFC that persists in adulthood. The mechanisms underlying THC-elicited mTOR activation following acute administration at the adult stage or chronic intake during adolescence also differ. Activation of mTOR induced by THC in adult mice is mainly mediated by $CB_1$ receptors located in GABAergic terminals, leading to an inhibition of GABAergic tone in the hippocampus. This in turn enhances pyramidal glutamatergic inputs and promotes NMDA receptor-dependent mTOR activation in pyramidal neurons (Puighermanal *et al*, 2009, 2013). Though such a mechanism might likewise contribute to the initial phase of prefrontal mTOR activation in mice exposed chronically to THC during adolescence, the activation of mTOR during the entire adolescent period clearly depends on $5-HT_6$ receptor activation (Fig 7). Indeed, it was not observed in $5-HT_6^{-/-}$ mice and was prevented by the administration of a $5-HT_6$ receptor *neutral antagonist* during the adolescence period, but not by the $5-HT_6$ receptor blockade at the adult stage. The latter observation indicates it might result from a non-physiological $5-HT_6$ receptor activation by endogenously released 5-HT rather than constitutive activity, which might be caused by $CB_1$ receptor-mediated decrease in GABA release and the disinhibition of 5-HT terminals (Fig 7) in the PFC.

The mechanism underlying the persistent activation of mTOR in adulthood remains more uncertain. It is conceivable that adolescent exposure to THC and the resulting non-physiological mTOR activation interfere with the maturation of the GABAergic system that undergoes refinement in the PFC until the end of adolescence (Kilb, 2012; Zamberletti *et al*, 2014). This might lead to a sustained increase in PFC serotonergic tone (Teissier *et al*, 2017) and, consequently, a persistent $5-HT_6$ receptor activation by endogenously released 5-HT.

A recent RNA-seq study in rats treated with THC during adolescence showed a decreased expression of Raptor mRNA 2 weeks after the THC treatment, while the level of mTOR mRNA was not affected (Miller *et al*, 2019). This suggests that the enhanced mTOR activity in THC-treated animals does not result from an increased expression of proteins of the mTOR complex 1 (mTORC1), but rather from an increase in mTOR catalytic activity, consistent with the observed increase in $Ser^{2448}$ phosphorylation state.

The sustained increase mTOR signaling in PFC of THC-treated mice plays a key role in the emergence of cognitive deficits in adulthood. Indeed, the cognitive deficits induced by THC intake during adolescence were abolished by the concomitant administration of either $5-HT_6$ receptor antagonists or rapamycin and were absent in the $5-HT_6^{-/-}$ mice. Notably, the same treatments (followed by the same washout period) administered in adulthood did not induce a prolonged rescue of these deficits. Furthermore, THC administration in adulthood did not induce such a prolonged activation of mTOR signaling and associated cognitive deficits. Collectively, these observations confirm that the adolescence is a period of vulnerability to

extrinsic risk factors such as drug abuse that irreversibly affect brain signaling processes crucial for cognitive functions (Higuera-Matas *et al*, 2015; Renard *et al*, 2016; Saravia *et al*, 2019).

The sustained activation of mTOR signaling, under the control of 5-HT$_6$ receptors, elicited in PFC by adolescent THC exposure might induce cognitive deficits in adulthood through different mechanisms. First, mTOR plays a critical role in various neurodevelopmental processes, including neuronal progenitor proliferation, neuronal migration, growth of dendrites and axons, and synaptogenesis (Bockaert & Marin, 2015). Corroborating these observations, a large body of evidence indicates a deleterious influence of aberrant mTOR signaling upon cognition in rodent models of neurodevelopmental disorders (Ehninger *et al*, 2008; Sharma *et al*, 2010; Ricciardi *et al*, 2011; Meffre *et al*, 2012; Troca-Marin *et al*, 2012; Huber *et al*, 2015), indicating that the deregulation of mTOR signaling in specific brain areas at critical developmental periods can compromise cognition later in life. Second, mTOR activation level finely controls dendritic spine pruning and shaping of synaptic connections at later stages of brain maturation but contrasting effects of mTOR overactivation upon spine density and morphology have been reported in autism models (Tavazoie *et al*, 2005; Tang *et al*, 2014). THC exposure during adolescence results in premature pruning of spines and protracted atrophy of distal apical trees associated with alteration of synaptic markers that might lead to a reduction in the complexity of pyramidal neurons and a reduced capacity for plasticity in neural circuits central for normal adult behavior (Rubino *et al*, 2015; Miller *et al*, 2019). The role of mTOR in structural abnormalities induced by adolescent exposure to THC remains to be established. Finally, a recent study revealed that exposure of adolescent rats to THC affects selective histone modifications that impact the expression of genes associated with synaptic plasticity. Again, changes in histone modifications are more widespread and pronounced after adolescent exposure than after adult exposure, suggesting specific adolescent vulnerability (Miller *et al*, 2019; Prini *et al*, 2018). As mTOR activity controls the phosphorylation state of proteins involved in histone post-translational modifications and chromatin remodeling (Citro *et al*, 2015; Zhang *et al*, 2017), the sustained mTOR activation in PFC of mice exposed to THC during adolescence might also contribute to some epigenetic mechanisms underlying the development of cognitive deficits (Prini *et al*, 2018).

Irrespective of the structural changes and the molecular mechanisms elicited by the sustained non-physiological mTOR activation induced by chronic THC consumption during adolescence, we showed that it induces a disruption in the excitatory–inhibitory balance in PFC. This alteration in the E/I balance, which is a landmark of neurodevelopmental disorders (Dani *et al*, 2005; Bateup *et al*, 2013; Berryer *et al*, 2016) such as schizophrenia or autism (Benes *et al*, 1991; Vogels & Abbott, 2009; Marin, 2012), persisted until adulthood and resulted from a decrease in GABAergic transmission and an increase in glutamatergic transmission. It might also result from the 5-HT$_6$ receptor-dependent overactivation of mTOR, as the blockade of this pathway during adolescence restored a normal E/I balance in adulthood in THC-injected mice.

In an effort to identify the molecular mechanisms underlying the disruption of the E/I balance following adolescent exposure to THC, we showed that this treatment induces an increase in HCN1 activity that leads to changes in intrinsic properties of PFC layer V

pyramidal neurons characterized by a more depolarized resting membrane potential and a lower firing threshold. Previous RNA-seq studies demonstrated a decrease in HCN1 mRNA level 2 weeks after the chronic administration of THC to adolescent rats, suggesting that the observed increase in HCN1 activity does not result from an increase in HCN1 channel expression (Miller *et al*, 2019) but is rather reminiscent of previous findings indicating that CB$_1$ receptor activation induces deficits in hippocampal LTP and spatial memory formation through HCN channels (Maroso *et al*, 2016). The signaling cascade underlying CB$_1$ receptor-mediated activation of $I_h$ in the hippocampus involves a c-Jun-N-terminal-kinase (JNK) that in turn increases cGMP formation through the activation of nitric oxide synthase (Maroso *et al*, 2016). In contrast, HCN1-dependent alterations of intrinsic properties of PFC pyramidal neurons elicited by administration of THC during adolescence were independent of cGMP, but rather involved a cAMP-dependent mechanism that might itself result from the sustained activation of adenylyl cyclase mediated by prefrontal 5-HT$_6$ receptors. Previous studies have also suggested that HCN1 channels regulate synaptic plasticity by dendritic integration of synaptic inputs to pyramidal neurons and thus play a role in learning and memory (Magee, 1999; Nolan *et al*, 2004). Indeed, alteration in HCN1 activity might also explain the impairment of electrically induced LTD observed at PFC layer I/V synapses in mice injected with THC during adolescence.

It is well accepted that HCN1 are important during neuronal development and network wiring (Bender & Baram, 2008). Developing neurons have to modulate both their intrinsic properties as well as their firing to constantly adapt to their changing environment. Factors that interfere with this process and alter HCN1 activity may have long-lasting deleterious effects that lead to network dysfunction underlying cognitive impairment observed in several brain pathologies (Chen *et al*, 2001; Brewster *et al*, 2002). For instance, an alteration in HCN1 activity has been found in mouse models of Rett syndrome (Mecp2$^{-/y}$ mice) (Balakrishnan & Mironov, 2018) and Fragile X syndrome (Fmr1$^{-/y}$ mice) (Brager *et al*, 2012) as well as in epilepsy (Dube *et al*, 2006).

In conclusion, the present study shows that a sustained, non-physiological mTOR activation under the control of 5-HT$_6$ receptors plays a key role in the alteration of intrinsic neuronal properties, E/I balance and synaptic plasticity in layer V pyramidal neurons of the PFC of adult mice exposed to cannabis during adolescence. It also shows that blocking 5-HT$_6$ receptor-elicited mTOR activation during this critical period of PFC maturation definitively prevents emergence of cognitive deficits in mice exposed to cannabis during adolescence, whereas its blockade in adulthood does not induce such a long-term pro-cognitive effect. It suggests that 5-HT$_6$ receptor antagonists, which recently failed in Phase III clinical trials as symptomatic treatment of cognitive symptoms in Alzheimer's disease (Atri *et al*, 2018; Khoury *et al*, 2018), might be repositioned as "disease-modifying" treatment to prevent emergence of cognitive deficits in adolescent cannabis abusers. Such a strategy based on early administration of 5-HT$_6$ receptor antagonists is certainly more relevant than mTOR blockade by pharmacological inhibitors, as it will specifically prevent the non-physiological activation of prefrontal mTOR without affecting physiological cerebral mTOR activity, which plays a key role in numerous physiological processes such as synaptic plasticity (Bockaert & Marin, 2015; Younts *et al*, 2016; Switon *et al*, 2017; Ryskalin *et al*, 2018). Accordingly, 5-HT$_6$

receptor antagonism will not reproduce the severe side effects induced by mTOR inhibitors such as rapamycin, which limit their clinical development for the treatment of psychiatric diseases. Given the deleterious influence of aberrant mTOR activation in several genetic forms of autism spectrum disorders (Ehninger et al, 2008; Ehninger & Silva, 2011; Huber et al, 2015; Sato, 2016; Ryskalin et al, 2018), administration of 5-HT$_6$ receptor antagonists as soon as adolescence might profitably be evaluated in autism. Finally, the recent progress in the identification of patients with high risk of transition to schizophrenia (Millan et al, 2016) suggests that such a strategy might also be extended to the more numerous population of patients with schizophrenia.

# Materials and Methods

### Animals

Wild-type male and female C57BL/6JRj mice were purchased from Janvier Laboratories. 5-HT$_6^{-/-}$ mice have a C57BL/6JRj background. Mice from both sexes were indifferently used in behavioral, electrophysiological, and biochemical experiments. Mice were received at PND 26 and housed under standardized conditions with a 12-h light/dark cycle, stable temperature (22 ± 1°C), controlled humidity (55 ± 10%), and free access to food and water. CB1$^{-/-}$ mice and their littermates (background C57BL/6JRj) were housed in a specific area under standardized conditions as described above. Animal husbandry and experimental procedures were performed in compliance with the animal use and care guidelines of the University of Montpellier, the French Agriculture Ministry, and the European Council Directive (86/609/EEC). All the experiments were conducted from PND 60 or from PND 90 depending on the experimental conditions.

### Drugs and treatments

Δ-9-tetrahydrocannabinol (THC) was purchased from THC Pharm. Picrotoxin (Sigma P1675), 6-Cyano-7-nitroquinoxaline-2,3-dione disodium salt hydrate (CNQX; Sigma C239), and N-ethyl-1,6-dihydro-1,2-dimethyl-6-(methylimino)-N-phenyl-4-pyrimidinamine hydrochloride (ZD7288; Sigma Z3777) were obtained from Sigma-Aldrich. 4-Iodo-N-[4-methoxy-3-(4-methylpiperazin-1-yl)-phenyl] benzene-sulfonamide (SB258585; Tocris #1961), tetrodotoxin (Tocris #1078) and D,L-2-Amino-5-phosphonopentanoic acid (D,L-AP5; Tocris #0106) were purchased from Tocris and rapamycin from LC Laboratories (R5000). THC was dissolved in 5% ethanol and 5% cremophor in NaCl. Vehicle groups received this solution. THC was injected daily (5 mg/kg, i.p.) between PNDs 30 and 45. SB258585 (injected at 2.5 mg/kg, i.p.), rapamycin (injected at 1.5 mg/kg, i.p.), and CPPQ (injected at 2.5 mg/kg, i.p.) were dissolved in 5% DMSO and 5% Tween-80 in NaCl. All mice received the same number of injections. Correspondingly, control mice were successively injected with the vehicle used for THC and the vehicle used for SB258585/Rapamycin and are referred as Veh/Veh on the figures. Likewise, mice treated with the vehicle used for THC and SB/Rapa were referred as Veh/SB and Veh/Rapa, respectively, while mice treated with THC and the vehicle used for SB/Rapa were both referred to THC/Veh.

### Slice preparation

Animals were anesthetized with isoflurane before sacrifice. Slices were performed as previously described (Barre et al, 2016; Berthoux et al, 2019). Brains were removed and rapidly transferred into ice-cold dissection buffer maintained in 5% CO$_2$/95% O$_2$ and containing 25 mM NaHCO$_3$ (Sigma S5761), 1.25 mM NaH$_2$PO$_4$ (Sigma S8282), 2.5 mM KCl (Sigma P3911), 0.5 mM CaCl$_2$ (Sigma C5080), 7 mM MgCl$_2$ (Sigma M2670), 25 mM glucose (Sigma G7021), 110 mM choline chloride (Sigma C1879), 11.6 mM ascorbic acid (Sigma A4034), and 3.1 mM pyruvic acid (Sigma P3256). Coronal brain slices (300 μm) were cut in ice-cold dissection buffer using a vibratome (Leica VT1200S). Slices were then transferred to artificial cerebrospinal fluid (aCSF, containing 118 mM NaCl, 2.5 mM KCl, 26.2 mM NaHCO$_3$, 1 mM NaH$_2$PO$_4$, 11 mM glucose, 1.3 mM MgCl$_2$, 2.4 mM CaCl$_2$, and maintained in 5% CO$_2$/95% O$_2$), at room temperature (22–25°C).

### Electrophysiological recordings

Patch recording pipettes (3–5 MΩ) were filled with intracellular solution (115 mM Cs-gluconate, 20 mM CsCl (Sigma C3011), 10 mM HEPES (Sigma H3375), 2.5 mM MgCl$_2$, 4 mM Na$_2$ATP (Sigma A2383), 0.4 mM NaGTP (Sigma G8877), 10 mM sodium phosphocreatine (Sigma P7936), and 0.6 mM EGTA (Sigma E4378), pH 7.3). Whole-cell recordings were obtained from layer V pyramidal neurons (300–400 μm from pial surface) of prelimbic PFC using a Multiclamp 700B amplifier (Axon Instruments) under an Axioscope2 microscope (Zeiss) equipped with infrared differential interference contrast optics. Data were filtered at 2 kHz and sampled at 10 kHz using Digidata 1440A (Molecular Devices) under the control of pClamp 10 (Axon Instruments). There were no significant differences in input or series resistance among groups. For miniature postsynaptic currents, the recording chamber was perfused with ACSF supplemented with 1 μM tetrodoxin at 22–25°C. mEPSC and mIPSC recordings were performed in the presence of 0.1 mM picrotoxin and 0.1 mM CNQX with 0.1 mM D, L-AP5, respectively. Currents were monitored for 10 min, and the quantification was made during the last 2 min of recording with Clampfit 10.2 (Axon Instruments).

Intrinsic electrophysiological properties were assessed by current-clamp recordings. Patch recording pipettes were filled with intracellular solution (120 mM K-gluconate, 10 mM KCl, 10 mM HEPES, 1.8 mM MgCl$_2$, 4 mM Na$_2$ATP, 0.3 mM NaGTP, 14 mM Na-phosphocreatine, 0.2 mM EGTA, pH 7.2). Resting membrane potential of each cell was measured immediately after the whole-cell patch formation. Action potentials (APs) were evoked by a ramp current injection with 10-pA increments for 2 ms, and only the first AP was used to determine the AP threshold and the minimal current necessary to induce firing (rheobase). The firing rate was recorded by a current injection at 150 pA for 250 ms.

Hyperpolarization-activated cation current ($I_h$) was measured as the voltage sag to a 500 ms hyperpolarizing current injection (50-pA increment from −400 pA to 0). Synaptic transmissions were blocked by adding 0.1 mM picrotoxin, 0.01 mM CNQX, and 0.1 mM D,L-AP5.

Electrically induced LTD was generated by a pairing protocol consisting in seven trains of 50 Hz stimuli (100 pulses per train),

delivered at 0.1 Hz, as previously described (Berthoux *et al*, 2019). Currents were monitored for 6 min before tetanic stimulation. Quantification of AMPA EPSC amplitude was made during the last 5-min period before tetanic stimulation and during the last 5-min period of recording, using Clampfit 10.2 (Axon Instruments).

### Analysis of mTOR activity in mouse PFC

Mice PFC were rapidly dissected and homogenized in ice-cold buffer containing 0.32 M sucrose, 10 mM HEPES, pH 7.4, and a cocktail of protease and phosphatase inhibitors (Roche). Homogenates were centrifuged at $1,000 \times g$ for 10 min to remove nuclei and large debris. Protein concentration in supernatants (PFC protein extracts) was determined by the bicinchoninic acid method. Proteins were resolved on 4–15% gradient gels (Bio-Rad) and transferred electrophoretically onto nitrocellulose membranes (Bio-Rad). Membranes were incubated in blocking buffer (Tris–HCl, 50 mM, pH 7.5; NaCl, 200 mM; Tween-20, 0.1%, and skimmed dried milk, 5%) for 1 h at room temperature and overnight with primary antibodies in blocking buffer: rabbit anti-phospho-S$^{2448}$ mTOR 1:1,000 (Cell Signaling #2971), rabbit anti-phospho-T$^{389}$ p70S6K 1:1,000 (Cell Signaling #9205), and mouse anti-panActin 1:2,000 (Lab Vision MS-1295-B). Membranes were then washed and incubated with horseradish peroxidase-conjugated anti-rabbit or anti-mouse antibodies (1:4,000 in blocking buffer, Millipore #12-348 and 12-349, respectively) for 1 h at room temperature. Immunoreactivity was detected with an enhanced chemiluminescence method (ECL detection reagent, GE Healthcare) using a ChemiDoc™ Touch Imaging System (Bio-Rad).

### Immunohistochemistry

Mice were anesthetized with pentobarbital (100 mg/kg, i.p.; CevaSanteAnimale) and transcardially perfused with 4% paraformaldehyde, 0.1 M sodium phosphate buffer (PBS, pH 7.4). Brains were post-fixed overnight in the same solution and stored at 4°C. Coronal slices (50 μm) were cut in the PFC with a vibratome (Leica VT1000S) and stored at −20°C in a cryoprotectant solution until being processed for immunohistochemistry. Briefly, brain sections were blocked in 1% Triton X-100, 10% goat serum, 0.1 M PBS for 2 h, and immunostained overnight with primary antibodies: rabbit anti-5-HT$_6$ receptor 1:500 (ab103016, Abcam), guinea-pig anti-CB$_1$ receptor 1:500 (Frontier Institute #ab2571593), mouse anti-PSD95 1:500 (NeuroMab #75-028), mouse anti-Bassoon 1:1,000 (Enzo Life Sciences, #ADI-VAM-P5003), mouse anti-GAD65 1:500 (Millipore #MAB351R), and rabbit anti-SERT 1:500 (Sigma #PC177L). Slices were then incubated with Alexa 594- or Alexa 488-conjugated goat anti-rabbit (1:1,000, Jackson immunoresearch), Alexa 488-conjugated anti-guinea-pig IgG (1:1,000, Jackson immunoresearch) or Alexa 594-conjugated goat anti-mouse (1,000, Jackson immunoresearch). Sections were mounted and were analyzed under a 40× oil-immersion objective using a confocal microscope (Carl Zeiss LSM 780). Briefly, for each image, we determined a precise ROI that was applied for all images. Then, we performed segmentation by thresholding to generate binary images from each selection using the « Otsu » auto-threshold function. For co-localization analysis, we used Mander's split coefficients to identify the fraction of CB$_1$ receptors or 5-HT$_6$ receptors that co-localizes with synaptic markers and the fraction of CB$_1$ that co-localizes with 5-HT$_6$ receptors using the macro « coloc 2 » from ImageJ.

### Behavioral analysis

#### Novel object recognition task

One week before the test, mice were extensively handled: The first 2 days, the operator put his hand on the home cage of the animals to familiarize the animals to his presence, and the following days, the animals were handled few minutes per day by the operator. Testing was carried out in a Plexiglas box placed in a dimly lit room with clearly visible contextual cues (black on white patterns) on the surrounding walls. Mice were habituated to the arena for 10 min per day, on days 1 and 2. On days 3, mice performed the novel object recognition task. The mice had a 10-min training session, a 30-min retention interval during which they were transferred back to the home cage, and a 10-min test session. The objects were plastic toys (approximately 3 cm width, 3 cm length, 5 cm height) and were cleaned with 10% ethanol between sessions. The experiments were video-recorded, and exploration times (nose in contact or sniffing at < 1 cm) were measured by a blinded observer. Mice with total time exploration of less than 3 s in test session were excluded. Discrimination indexes [(exploration time of novel object − exploration time of familiar object)/total object exploration time] were compared between groups.

#### Social preference task

Testing was carried out in a rectangular, three-chamber box with dividing walls made of clear Plexiglas and an open middle section (60 cm length × 40 cm width in total, each compartment has a length of 20 cm), which allows free access to each chamber (Kaidanovich-Beilin *et al*, 2011). Before the test, each tested mouse was placed for 10 min in the middle compartment a three-chamber device, without the walls between the compartments, to allow free access to the three compartments. Then, a control mouse (same age, same sex as tested mice) was placed inside a wire containment jail, located in one chamber. In the second chamber, an object was placed inside a wire containment jail. Then, the walls were removed between the compartments, to allow free access for the test mouse to explore each of the three chambers. Duration of direct contacts between the tested mouse and the containment jail or cup housing the conspecific or object was recorded for 10 min. The experiments were video-recorded, and exploration times (nose in contact or sniffing at < 1 cm) were measured by a blinded observer. Sociability indexes [(exploration time of congener − exploration time of object)/total exploration time] were compared between groups.

#### Social discrimination task

For the social novelty discrimination test (ability of an adult mouse to discriminate novel from familiar congeners), the same procedure as for the social preference task was used but the object in the wire containment jail was replaced by a mouse (same age, same sex as the tested mouse) during the second session. The walls between the compartments were then removed to allow free access to each of the three chambers for the tested mouse. The time of interaction the test mice with each congener was recorded for 10 min. Mice with total time exploration of less than 3 s in test session were excluded.

Social discrimination indexes [(exploration time of the novel congener − exploration time of familiar congener)/total exploration time] were compared between groups.

### Elevated Plus Maze

Anxiety-related behavior was assessed in an Elevated Plus Maze (EPM) consisting in a black plexiglass apparatus with four elevated (50 cm above the floor) arms (26 cm long × 5 cm wide) set in cross from a neutral central square (5 × 5 cm). Two opposing arms, chipboard walls, and two opposing arms were devoid of walls (open arms), under dim lighting conditions (> 50 lux). Cumulative time spent in open (aversive) and closed (non-aversive) arms was recorded during a 5-min session.

### Circular corridor

Locomotor activity was measured for 60 min in a circular corridor (14 cm wide, 18 cm in diameter) with four infrared beams placed at 90° angles (Imetronic, Pessac, France) in a low luminosity environ.

### Statistical data analysis

For biochemistry, immunohistochemistry, and electrophysiology, a minimum of three animals was used in each group. Mice from both sexes were indifferently used, and they were submitted to behavioral experiments in a randomized sequential order. Behavioral recordings were blindly analyzed. Data were analyzed using GraphPad Prism software (v. 7.0). The homogeneity of sample variance was tested using Brown–Forsythe's test. As long as no variance among groups was detected, ANOVA was performed. Statistical significance was determined by one-way ANOVA followed by Newman–Keuls test for electrophysiological and biochemical experiments, or by one-way ANOVA followed by Bonferroni test for behavioral experiments. Detailed statistics for figures are in the Appendix Table S1.

**Expanded View** for this article is available online.

## Acknowledgements

We wish to thank Dr Laurent Fagni for critical reading of the manuscript and the animal facility's staffs of the Institute of Functional Genomics for the daily care of animals, especially Steeve Thirard and Denis Greuet. This study was supported by grants from CNRS, INSERM, University of Montpellier, Labex EpiGenMed, Fondation pour la Recherche Médicale (FRM, "Physiopathologie de l'Addiction program", contract no DPA20140629800), Fondation FondaMental and ANR (Contract no 17-CE16-0013-01). C. Berthoux was a recipient of a fellowship from the Gouvernement de la Nouvelle Calédonie.

## Author contributions

CBer performed electrophysiology experiments. CBer, CC and AR performed biochemistry experiments; AMH designed, performed and analyzed behavioral experiments. AR and ELD performed and analyzed behavioral experiments; AR and FA performed immunohistochemistry. PZ and KG synthesized and provided CPPQ. SCD provided the 5-HT$_6^{-/-}$ mouse cohort. RM provided the CB1$^{-/-}$ mouse cohort. CBéc, JB and PM designed the experiments and supervised the study; CBéc, PM and CBer wrote the manuscript.

## Conflict of interest

The authors declare that they have no conflict of interest.

### The paper explained

#### Problem

Cannabis is the most commonly abused illicit drug. Epidemiological studies suggest that cannabis abuse during adolescence confers an increased risk for developing later in life psychotic-like symptoms and cognitive deficits reminiscent to those observed in schizophrenia, suggesting common pathological mechanisms that remain poorly characterized. It is conceivable that cannabis consumption might interfere with maturational events occurring in the adolescent brain, leading to alterations of brain connectivity and functionality similar to those observed in individuals with schizophrenia. Due to the huge increase in the number of adolescents regularly consuming cannabis, at younger and younger age, and the increase in the concentration of Δ9-tetrahydrocannabinol (THC) in currently used cannabis, there is an urgent need of innovative therapies for the clinical management and, ultimately, the prevention of psychotic and cognitive symptoms occurring in adulthood in adolescent cannabis abusers.

#### Results

In line with previous findings that revealed a role of mTOR activation, under the control of serotonin 5-HT$_6$ receptor, in cognitive deficits in rodent neurodevelopmental models of schizophrenia, we show that chronic administration of THC to mice during adolescence induces a long-lasting activation of mTOR in the prefrontal cortex (PFC) that persisted in adulthood. This sustained activation of mTOR, otherwise absent in 5-HT$_6$ receptor-deficient mice, was prevented by concomitant administration of 5-HT$_6$ receptor antagonists. THC administration during adolescence also affected excitatory and inhibitory transmission in PFC, intrinsic properties of layer V pyramidal neurons, and long-term depression at PFC layer I/V synapses, and induced deficits in social behaviors and cognition in adulthood. The synaptic and neuronal alterations as well as the associated cognitive deficits were also prevented by the blockade of 5-HT$_6$ receptor-operated mTOR signaling pathway during adolescence. In contrast, they were still present after the same treatments delivered at the adult stage.

#### Impact

These observations suggest a role of 5-HT$_6$ receptor-operated mTOR signaling in abnormalities of cortical network wiring elicited by THC at a critical period of brain maturation. They also suggest 5-HT$_6$ receptor antagonists that are well tolerated and still in clinical evaluation as cognitive enhancers in dementia might be repositioned as early treatment to prevent the emergence of cognitive deficits at the adult stage in adolescent cannabis abusers.

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
