## [Review Process File · EMBO Molecular Medicine]

Early 5-HT₆ receptor blockade prevents symptom onset in a model of adolescent cannabis abuse

Coralie Berthoux, Al Mahdy Hamieh, Angelina Rogliardo, Emilie L. Doucet, Camille Coudert, Fabrice Ango, Katarzyna Grychowska, Séverine-Chaumont-Dubel, Pawel Zajdel, Rafael Maldonado, Joël Bockaert, Philippe Marin and Carine Bécamel

Review timeline:

Submission date:	15th Mar 2019
Editorial Decision:	26th Apr 2019
Revision received:	30th Oct 2019
Editorial Decision:	21st Nov 2019
Revision received:	9th Jan 2020
Editorial Decision:	12th Feb 2020
Revision received:	5th Mar 2020
Accepted:	10th Mar 2020

Editor: Céline Carret

Transaction Report:

1st Editorial Decision

26th Apr 2019

Thank you for the submission of your manuscript to EMBO Molecular Medicine. We have now heard back from the three referees whom we asked to evaluate your manuscript.

You will see that while two referees find the study of significant interest, referee 1 is more reserved and offers several suggestions that if followed, would strongly strengthen the study. Of particular importance for our scope and aims, and given the somehow limited advance of the findings and unclear translational relevance at this point, mechanistic insights should be added to increase the conclusiveness the findings. It is our opinion that all suggested experiments and text modifications are reasonable and would improve the impact of the paper and I would therefore encourage you to address these in a major revision of your work.

Please note that EMBO Molecular Medicine strongly supports a single round of revision and that, as acceptance or rejection of the manuscript will depend on another round of review, your responses should be as complete as possible.

***** Reviewer's comments *****

Referee #1 (Comments on Novelty/Model System for Author):

I do not think the paper provides any mechanistic information that could be used to assure the adequacy of the model system. In fact, this is a purely phenomenological study. Given the lack of experimental detail (see below), I doubt these results stand the chance to directly enter "medicine" even though the general idea seems appealing.

Referee #1 (Remarks for Author):

This study addresses if 5HT6 receptor modulation during adolescence can normalize THC-induced behavioral and electrophysiological (and biochemical) changes, particularly in the mouse prefrontal cortex. A set of descriptive experiments is produced that all seem to have worked as one would ideally expect, even if their design or outcome seemingly contradict basic principles of pharmacology or circuit neurobiology.

Major:

Individually, these experiments tell nothing new about THC/cannabis and adolescence since each experiment was performed by others previously (mTOR, adolescent electrophysiology, HCN1). Bringing in 5HT6 receptor modulation seems to be the "wonder treatment" to normalize everything, even though neither detailed pharmacology (dose-response relationships) nor the discovery of molecular/cellular drug substrates is given. A major caveat here is that the authors do not attempt to reconcile THC action (presynaptic CB1R dependent? receptor independent?) with likely postsynaptic 5HT6 receptor activity. CB1Rs are Gi coupled (mostly) while 5HT6 receptors signal through Gs. Therefore, one would wonder how THC reduces resting membrane potential and 5HT6 re-hyperpolarize neurons.

Is there a PFC neuron (eg pyramidal cell) that expresses both receptors (i.e. signal "hub" concept) or are the two actions entirely cross-correlative and parallel (subcortical 5HT afferents - one pathway, THC action within cortex - another pathway). Thus, the paper lacks sufficient novelty and, most unfortunately, experimental rigorous to be evaluated positively.

For causality, the authors should use genetic models (KOs for both THC targets and 5HT6 receptors), minimally. In a way, if their electrophysiology and HCN1 pharmacology data were correct, placing a DREADD/optogenetic construct that hyperpolarizes PFC neurons shall equally rescue THC phenotype, thus rendering the 5HT6 component of this study dispensable (and any other channel could do the job).

Rudimentary electrophysiology is insufficient here. There is a need to test plasticity and circuit mechanisms. Where does the defunct GABA input originate from?

It is also a very unorthodox idea that long-term maladaptation can acutely be (and completely) reversed by a HCN1 antagonist. Then, all kids who smoked cannabis could be "cured for life", which does not seem to be the case. In addition, the HCN1 context does not integrate the observation of sustained mTOR activation, again allowing for an at least binary view on the data shown here.

Coming from this, how do the authors explain the sustained and site-directed phosphorylation of mTOR? If they were correct then a kinase upstream of mTOR needs to be constitutively activated. Did the authors exclude this being an artifact? If correct, what is the mechanism of this primary step.

More minor:

The title is misleading, because there is no human data presented here. The fact this is a model study shall be clear from the outset.

The paradigm of delayed 5HT6 antagonism in adult with THC in adolescence is totally contradictory to the introduction, which is pitched to say that only adolescence THC has life-long effects because of interference with developmental processes.

Yasmin Hurd's group published a wealth of data on adolescence and THC action, most recently using single-cell RNA-seq. Do their data (which can publicly be reprocessed) match any of the observations (mTOR, HCN1 etc) here?

Precise description of the drug interaction studies is needed.

Cannabis and THC are not interchangeable. Being restricted to "THC" throughout is a must.

"In conclusion, the present study shows that a sustained, non-physiological mTOR activation under

the control of 5-HT₆ receptors, plays a key role in the alteration of intrinsic neuronal properties and E/I balance in the PFC of adult mice exposed to cannabis during adolescence" seem to be an overstatement and unjustified by the lack of molecular, cellular or circuit specific data.

How were drug concentrations selected? This is particularly pertinent to THC interaction studies which returned negative data (e.g. on page 7) to prove that the experiments are not dose-biased.

P6: what is the discrimination index of vehicle-treated animals? Moreover, data on THC alone must be added/described, too.

Figures are inadequate. Representative diagrams, traces must be shown to illustrate each point exhaustively.

Referee #2 (Comments on Novelty/Model System for Author):

This is an interesting study demonstrating a new pharmacological target to counteract the long-term negative cognitive effects of cannabis use in adolescence.

Authors previously showed that 5-HT₆ receptors physically interact with mTOR to increase mTOR signaling and that 5HT₆ receptor antagonists could reverse cognitive and social interaction defects in schizophrenia mouse models (Meffre et al. 2012). In the present study they carry this analysis further focusing on another neurodevelopmental disorder: cannabis use during adolescence also known to increase mTOR signaling. THC administration from P30 to P45 increased mTOR-phosphorylation in the PFC and caused cognitive alterations at P60 (novel object recognition test and social interaction test). These effects were annulled by 5-HT₆ antagonist co-administered with THC, but not when the 5HT₆ antagonist were administered 15 days later. Additionally, authors provide some mechanistic insights on the cellular underpinings of these long term changes induced by THC. Using electrophysiological recordings of PFC slices, they show that adolescent THC reduces GABA input, and enhances excitatory drive onto layer V neurons. These neurons also show increase in their intrinsic excitability, that are reversed by a HCN1 channel antagonist. These effects are also dependent on the 5HT₆R and appear to be downstream of mTOR signaling .

The paper is clearly written, well illustrated with adequate controls and contributes to better understand long-term effects of cannabis in adolescents , and proposes potentially important a new pharmacological target to reverse these effects.

My criticism are minor and essentially suggestions to clarify some points.

- 1) Authors state that HT₆ antagonists have no effect to reverse behaviour and mTOR in later administration protocols. However the time frames of the experiments are not the same. Authors state that mTOR activation is persistent throughout life, however comparison of figures 1 and 2, shows that mTOR phosphorylation could decrease from P60 to P90, suggesting that it gradually wear off. Please comment.
- 2) On the same line : have the authors tried to administer THC from P60 to P75 to determine whether effects are really restricted to the juvenile period.
- 3) Ephys data, suggest a reduced GABA input on layer 5 PFC neurons, and enhanced HCN1 channel expression. It would be nice if the authors could backup this interesting lead with morphology, or gene expression studies. Particularly since these developmental mechanisms are discussed
- 4) Rapid reversal of hyperexcitability was obtained with the HCN1 channel blocker ZD7288 on PFC slice. Can this compound be administered in vivo to test whether this might reverse behavioural abnormalities; This seemed like an obvious experiment, but maybe not possible.
- 5) Discussion on the possible developmental impact of enhanced mTOR signalling under the control of 5-H₆R seemed overly long and speculative since the authors do not provide new insight on these mechanisms

Referee #3 (Remarks for Author):

In this study, Berthoux and colleagues investigate the long-term effects of chronic THC use in adolescent mice. In particular, they show that THC daily injections from Postnatal Day (P) 30 to P54 lead to

- increased mTOR activation in prefrontal cortex (but not hippocampus)
- impaired behavior in the novel recognition task (THC-treated mice failed to recognize the novel object) and in the social preference task (THC - treated mice spend less time with the mouse compared to vehicle treated mice)
- altered mIPSC (reduced) and mEPSC (increased) frequency in L5 pyramidal neurons
- increased sag (due to HCN1 channel activity), leading to increased resting membrane potential and action potential threshold in L5 pyramidal neurons.

Remarkably, simultaneous treatment during adolescence with rapamycin (mTOR inhibitor) or SB285585 (5HT6 inhibitor) rescues the biochemical, electrophysiological and behavioral phenotypes. Conversely, two weeks treatment with SB or RAPA from P60-75 does not normalize mTOR hyperactivation or deficits in novel object recognition.

This is a very timely, very well written and well-executed study, which sheds light on an important issue, namely the potential molecular mechanisms underlying the long-term effect chronic THC use in adolescence. I have relatively minor comments and suggestion that would help readers to evaluate and reproduce the data.

1) In methods section, the authors describe the two different vehicle solution used for THC and SB/RAPA. Are data regarding mice treated with THC and with the vehicle for RAPA/SB shown? Are data regarding mice treated with the vehicle for RAPA/SB alone? I do not think so. Please, correct methods accordingly by clarifying that the only control used are mice treated with the vehicle for THC alone or vehicle for THC+SB or vehicle for THC+RAPA.

2) The authors recorded 5 neurons from prelimbic or anterior cingulate cortex. Why were two regions chosen? Were n of recorded cells equally taken from these two regions? Are electrophys properties comparable? Please, add this information in the methods.

3) In the methods/novel object recognition task, the authors write that 'Mice were extensively handled and habituated to the test'. Please, be more precise so other operators can reproduce the experiments. Authors also write that novel object recognition task was performed on day 3 and 4. What was the difference in the tests performed on day 3 and 4? Were completely different objects used? Where discrimination index combined on day 3 and 4?

4) In the methods/social preference task. Were mice habituated to the chamber? If yes, for how long? How were cogenitor mice chosen (what was the age range, for example).

5) Were the same mice tested for novel object recognition and social preference? If yes, how long did operators waited between the two tests?

6) Where mice of both sexes tested in behavior and in all other experiments? Add this important information.

7) In the methods section, please add all catalog numbers of antibodies and other critical reagents.

8) Results section. As a general comment, please add exact N of mice or n of recorded cell/ N of mice for each experimental group. It would be helpful to plot all data set as it was done for the behavioral data, namely showing all data point and not only average values.

9) Figure 3. N of recorded neurons is missing. Please see comment #6

10) Figure 4. Please add exact n from exact N of mice, per groups.

11) Fig 2 and 3. It would be helpful to plot raw values for all data groups and not only values

normalized on vehicle.

12) Supplemental Figures S1, S2, S3 and S4. Please write that mice were treated with Veh alone or Veh+SB or Veh+RAPA. Figure S3, n of recorded cells in each experimental group is missing. Figure S4, n of cells and N of mice is missing. S5, n of recorded cells is missing. Please see and apply comment #6 to all supplemental figures, too.

13) Is locomotion and anxiety behavior similar in vehicle and THC-treated mice? These parameters can be evaluated using open field and elevated plus maze tests and are critical to interpret altered social behavior.

14) Data showing whether the activity of the HCN1 channels is rescued by RAPA or SB treatment during adolescence would further strengthen the authors' conclusions.

15) Figure 2. It would be interesting, if possible, to add data for social discrimination too, for consistency.

1st Revision - authors' response

30th Oct 2019

Points raised by Reviewer 1:

Major:

1) Individually, these experiments tell nothing new about THC/cannabis and adolescence since each experiment was performed by others previously (mTOR, adolescent electrophysiology, HCN1). Bringing in 5HT6 receptor modulation seems to be the "wonder treatment" to normalize everything, even though neither detailed pharmacology (dose-response relationships) nor the discovery of molecular/cellular drug substrates is given. A major caveat here is that the authors do not attempt to reconcile THC action (presynaptic CB1R dependent? receptor independent?) with likely postsynaptic 5HT6 receptor activity. CB1Rs are Gi coupled (mostly) while 5HT6 receptors signal through Gs. Therefore, one would wonder how THC reduces resting membrane potential and 5HT6 re-hyperpolarize neurons.

We acknowledge that several studies have demonstrated a role of mTOR signaling in the amnesic effects induced by an acute administration of THC at the adult stage (Puighermanal et al. *Nature Neuroscience*, 2009, 12(9):1152-8) and the modulation of HCN1 channel by CB₁ receptor activation (Maroso et al. 2016, *Neuron*, 89:1059-1073). However, to our knowledge, no study has currently investigated the implication of those mechanisms *in the context of chronic consumption of THC during adolescence*. Strikingly, we show that chronic THC consumption during adolescence induces a sustained activation of mTOR in the prefrontal cortex, which persists 15 days after the last THC injection (Figure 1 and page 6 of the manuscript), in contrast with the corresponding treatment performed at the adult stage (Supplementary Figure S6 and page 9 of the manuscript). These findings led us to hypothesize that adolescent THC intake might interfere with network shaping at a critical stage of prefrontal cortex maturation. This line of thinking provided the impetus to explore the possible influence of the 5-HT₆ receptor as i) this receptor, like mTOR, has been identified as a key regulator of neuronal differentiation and thereby might strongly influence brain circuit wiring and ii) a non-physiological mTOR activation in the prefrontal cortex, under the control of 5-HT₆ receptors, has been involved in cognitive deficits observed in adulthood in neuro-developmental models of schizophrenia, which share common features with those observed in adolescent cannabis abusers. We feel that the demonstration of a role of the 5-HT₆ receptor-mTOR pathway and of the resulting network dysfunction that leads to the alteration of HCN1 channel activity provides by itself a significant advance into the mechanism underlying the cognitive symptoms induced by chronic THC intake during adolescence.

We did not claim that 5HT₆ receptor modulation is the wonder treatment that normalizes everything in THC-treated animals. We provide a *proof of concept* demonstrating the

potential of 5-HT₆ receptor antagonists (otherwise still under clinical evaluation for other applications), administered at a specific period of the prefrontal cortex maturation, to prevent cognitive deficits and synaptic alterations in a model of chronic consumption of THC during adolescence.

We used single doses of THC, the 5-HT₆ receptor antagonist and rapamycin due the complexity and duration of the protocol involving chronic treatments, and the large cohort size requested. Doses were chosen based on our previously published work using these compounds *in vivo* (see point 13).

As outlined below, the specificity of the effect of THC and the 5-HT₆ receptor antagonist is now further demonstrated by the use of 5-HT₆ and CB₁ receptor knock-out mice. We also provide data indicating that both receptors are located in different neurons in the prefrontal cortex, ruling out a crosstalk (or a signaling hub) between receptor-operated signaling within a given neuronal population.

2) Is there a PFC neuron (eg pyramidal cell) that expresses both receptors (i.e. signal "hub" concept) or are the two actions entirely cross-correlative and parallel (subcortical 5HT afferents - one pathway, THC action within cortex - another pathway).

We have added an immunohistochemistry experiment on prefrontal cortex slices which indicates that 5-HT₆ and CB₁ receptors are not co-localized in the same neurons: whereas 5-HT₆ receptor immunoreactivity was found in the soma plasma membrane of some pyramidal neurons, CB₁ receptors were mainly detected on fibres and nerve terminals (presumably arising from CB₁ receptor-expressing interneurons), consistent with previous findings (Eggan et al., 2007, Cereb Cortex 17, 175-191; Cathel et al., 2014, Eur J Neurosci 40, 3202-3214). Moreover, some of CB₁-positive fibres enwrapping somas stained with the 5-HT₆ receptor antibody (Supplementary Figure S4 and page 7 of the manuscript). This suggests that THC-induced mTOR overactivation is rather due to the regulation of local network activity or network rearrangement rather than signaling cross-talks in neurons co-expressing both receptors.

Thus, the paper lacks sufficient novelty and, most unfortunately, experimental rigorous to be evaluated positively.

We agree with the reviewer that the mechanism underlying the beneficial effects of blocking the 5-HT₆-mTOR pathway remains partially characterized. As mentioned in Point 1, we hypothesize that THC administration might interfere with maturational events occurring in the adolescent prefrontal cortex that might lead to the rearrangement of cortical networks affecting the serotonergic system itself. We now provide results suggesting that the sustained activation of 5-HT₆ receptors in THC-treated mice might result from an increase in serotonergic tone rather than from the enhancement of receptor's constitutive activity: the effect of SB258585, which behaves as inverse agonist at 5-HT₆ receptors (Meffre et al. EMBO Mol. Med., 4:1043-1056, 2012), was reproduced by the injection of the recently characterized neutral antagonist CPPQ at 5-HT₆ receptors (Deraredj-Nadim et al. 2016, PNAS, 113:12310-12315) (Supplementary Figure S2B and page 6 of the manuscript).

The activation of 5-HT₆ receptors seems to be responsible for the alteration of HCN1 activity, as injecting CPPQ to THC-treated mice during adolescence restored normal resting membrane potential, AP threshold and rheobase in pyramidal neurons (see Supplementary Figure S9 and page 10 of the manuscript). 5-HT₆ receptors are canonically coupled to Gs and activate adenylyl cyclase in addition to mTOR pathway. As HCN1 is known to be modulated by cyclic nucleotides, we also examined the effect of an adenylyl cyclase inhibitor, DDOA, on intrinsic properties of pyramidal neurons and showed that perfusion of DDOA into the recording pipette restored normal intrinsic neuronal properties (see Figure 5 and "page 12 of the manuscript"). In contrast, perfusion of ODQ into the recording pipette, an inhibitor guanylate cyclase (otherwise activated upon CB₁ receptor stimulation) did not restore normal intrinsic properties of pyramidal cortical neurons (see Figure 5 and "page 12 of the manuscript").

Collectively, these novel and previously described results suggest that chronic THC injection during adolescence might lead to an increase in PFC serotonergic tone and consequently, to 5-HT₆ receptor over-activation which in turn might result in enhanced

cAMP production, and cAMP-dependent alteration of HCN1 channel activity and neuronal intrinsic properties.

3) For causality, the authors should use genetic models (KOs for both THC targets and 5HT₆ receptors), minimally. In a way, if their electrophysiology and HCN1 pharmacology data were correct, placing a DREADD/optogenetic construct that hyperpolarizes PFC neurons shall equally rescue THC phenotype, thus rendering the 5HT₆ component of this study dispensable (and any other channel could do the job).

We thank the reviewer for his valuable suggestion, and we acknowledge that adding experiments in 5-HT₆^{-/-} mice is an important point to confirm the effects of 5-HT₆ antagonists. Now, we show that administration of THC to adolescent 5-HT₆^{-/-} mice does not induce an activation of mTOR (assessed by measuring mTOR phosphorylation at Ser₂₄₄₈ and p70S6K phosphorylation at Thr₃₈₉) in the prefrontal cortex at the adult stage. Corroborating these biochemical results, THC administration during adolescence did not affect novelty discrimination in the novel recognition test nor the sociability of 5-HT₆^{-/-} mice. These new results (illustrated on Figure 1D and E and described in "Results", page 6) confirm the data obtained in wild type mice treated with 5-HT₆ receptor antagonists and irrefutably demonstrate the role of 5-HT₆ receptors in cognitive deficits induced by THC intake during adolescence.

Though more expected, we also provide results indicating that THC administration does not induce mTOR action in CB₁^{-/-} mice, confirming the role of CB₁ receptors in the THC effects (see Supplementary Fig S2A and "Results" page 6 of the manuscripts).

We agree with the reviewer that placing DREADD/optogenetic constructs to hyperpolarizes prefrontal cortex neurons would provide more information, but those approaches remain extremely challenging to implement in adolescent mice. Moreover, using DREADD constructs to shut-down pyramidal neurons during the critical period of adolescence could induce additional deleterious and persistent neuronal network rearrangements. The precise goal of our study was not to rescue the THC-induced phenotypes by manipulating any channel such as HCN channels, thus rendering 5-HT₆ receptor dispensable, but rather to manipulate the 5-HT₆-mTOR pathway thanks to 5-HT₆ receptor antagonists which have proven their safety and are currently in development for other applications.

4) Rudimentary electrophysiology is insufficient here. There is a need to test plasticity and circuit mechanisms. Where does the defunct GABA input originate from?

The electrophysiology experiments described in the initial version are not "rudimentary experiments", they clearly established that blocking the 5-HT₆-mTOR pathway prevents the alteration of intrinsic properties of prefrontal cortex pyramidal neurons and the disruption in excitatory/inhibitory balance induced by THC administration during adolescence, which both reflect a dysregulation of the entire cortical network connectivity.

As requested by the reviewer, we have performed LTD experiments. Corroborating previous findings (Rubino *et al.* Neurobiol. Dis., 73:60-9, 2015), we show that THC administration during adolescence impairs LTD at PFC layer I/V synapses. Furthermore, normal LTD was restored in mice treated with the 5-HT₆ receptor antagonist SB258585 during adolescence (see Figure 6 and "Results", page 12-13).

A previous study has shown that adolescent THC exposure results in reduced GAD67 level in parvalbumin- and cholecystinin-containing neurons, suggesting that both GABAergic interneuron subpopulations might play a role in THC-induced symptoms (Zamberletti *et al.* Neurobiol. Dis., 63:35-43, 2014). This point has now been added in the discussion, Page 16.

5) It is also a very unorthodox idea that long-term maladaptation can acutely be (and completely) reversed by a HCN1 antagonist. Then, all kids who smoked cannabis could be "cured for life", which does not seem to be the case. In addition, the HCN1 context does not integrate the observation of sustained mTOR activation, again allowing for an at least binary view on the data shown here.

The HCN1 inhibitor ZD7288 was acutely applied to prefrontal cortex slices to demonstrate the role of HCN1 channel in the observed changes in the intrinsic neuronal properties, not to reverse long-term maladaptation in THC-treated mice. This compound likely induces

deleterious effects *in vivo* because HCN channels are also expressed in the periphery, especially in the heart where ZD7288 reduces heart rate (Luo *et al.* Neuroscience, 144(4):1477-85, 2007).

As previously mentioned, alteration of HCN1 channel activity in THC-treated mice might result from the sustained activation of 5-HT₆ receptor as a consequence of altered network activity, leading to an increase in cAMP production and cAMP-dependent alteration of HCN1 channel function (see Point 2).

6) Coming from this, how do the authors explain the sustained and site-directed phosphorylation of mTOR? If they were correct, then a kinase upstream of mTOR needs to be constitutively activated. Did the authors exclude this being an artefact? If correct, what is the mechanism of this primary step.

We previously demonstrated that 5-HT₆ receptor-mediated mTOR activation depends on both the physical interaction of the receptor C-terminal domain with mTOR and the canonical PI3K/Akt/Rheb pathway (Meffre *et al.* EMBO Mol. Med., 4:1043-1056, 2012).

This is likely not an artefact as 5-HT₆ receptor-dependent mTOR activation has been described in two other more recent studies (Wang *et al.*, Mol Neurobiol. 51:1292-9, 2015; Teng *et al.* Plos Biol. 17(3):e2007097, 2019). We wish to stress that this pathway has now been involved in seizure activity in epilepsy and structural alterations induced by dietary restriction, suggesting that it influences several pathophysiological processes in addition to cognitive deficits observed in preclinical models of schizophrenia and adolescent THC consumption.

More minor:

7) The title is misleading, because there is no human data presented here. The fact this is a model study shall be clear from the outset.

We agree with the reviewer's comment and modified the title, which now refers to "a model of adolescent cannabis abuse".

8) The paradigm of delayed 5HT6 antagonism in adult with THC in adolescence is totally contradictory to the introduction, which is pitched to say that only adolescence THC has life-long effects because of interference with developmental processes.

We included a control experiment where adult mice were treated with the 5-HT₆ antagonist or rapamycin (Figure 2) only to demonstrate that blocking the 5-HT₆-mTOR pathway at the adult stage does not induce long-term beneficial effects, in contrast to the corresponding treatments performed during adolescence.

Furthermore, to confirm that THC induces life-long effects due to interference with developmental processes, we also injected with THC at the adulthood (from P60 to P75) as a control experiment. These mice exhibited similar performance in the novel object recognition task, assessed 15 days after the last THC injection to mimic the protocol applied to animals injected with THC during adolescence, as vehicle-injected animals (Figure S6 and page 9 of the manuscript).

9) Yasmin Hurd's group published a wealth of data on adolescence and THC action, most recently using single-cell RNA-seq. Do their data (which can publicly be reprocessed) match any of the observations (mTOR, HCN1 etc) here?

We thank the reviewer for his recommendation. We checked recent RNA-seq data obtained by Yasmin Hurd's group in rats treated with THC during adolescence (Miller *et al.*, Mol. Psy., 24:588-600, 2019) and focused on genes regulated in THC- vs. vehicle-treated animals two weeks after the treatment to better match with our protocol. Whereas mTOR was not affected, a decrease in Raptor and HCN1 mRNA levels was found in THC-treated rats. 5-HT₆ receptor mRNA was not referenced in this study. These findings indicate that the overactivation of mTOR and the resulting dysregulation of HCN1 channel are not caused by an increase in their gene expression but rather from an increase in their catalytic (mTOR) or channel activity and are now discussed on page 17.

10) Precise description of the drug interaction studies is needed.

Our study only provides a proof of concept that 5-HT₆ receptor antagonists, administered at a specific period of brain maturation, prevent cognitive deficits in a *preclinical model* of cannabis consumption during adolescence. Although we agree with the reviewer that a precise description of the drug interaction is needed before the clinical validation of the concept, it is not mandatory at this stage.

11) Cannabis and THC are not interchangeable. Being restricted to "THC" throughout is a must.

We agree with the reviewer's comment. We replaced "cannabis" by "THC" throughout the manuscript when it was relevant.

12) "In conclusion, the present study shows that a sustained, non-physiological mTOR activation under the control of 5-HT₆ receptors, plays a key role in the alteration of intrinsic neuronal properties and E/I balance in the PFC of adult mice exposed to cannabis during adolescence" seem to be an overstatement and unjustified by the lack of molecular, cellular or circuit specific data.

We hope that the addition of novel data, especially those performed on 5-HT₆^{-/-} mice, will convince the reviewer that this conclusion is not an overstatement.

13) How were drug concentrations selected? This is particularly pertinent to THC interaction studies which returned negative data (e.g. on page 7) to prove that the experiments are not dose-biased

The THC dose selected (5 mg/kg) corresponds to two joints per day according to the transformation in human equivalent dose proposed by the FDA. This THC dose is slightly higher than the minimal dose (3 mg/kg) that induces an activation of mTOR signaling associated with significant memory impairment in our previous study (Puighermanal *et al.* Nature Neurosci., 12:1152-8., 2009).

The SB258585 dose selected (2.5 mg/kg) corresponds to the dose that normalizes mTOR activation levels and rescues cognitive deficits in neurodevelopmental models of schizophrenia (Meffre *et al.* EMBO Mol. Med., 4:1043-1056, 2012).

The rapamycin dose selected (1.5 mg/kg) corresponds to the dose used for sub-chronic administration and that rescued the deficit in novel object recognition task in rats treated with THC (Puighermanal *et al.* Nature Neurosci., 12:1152-8, 2009).

14) P6: what is the discrimination index of vehicle-treated animals? Moreover, data on THC alone must be added/described, too.

The discrimination index of vehicle-treated animals has been added in the text (see "Results, page 7).

15) Figures are inadequate. Representative diagrams, traces must be shown to illustrate each point exhaustively.

Figures have been redrawn and now show each point exhaustively, as requested.

Referee #2 (Comments on Novelty/Model System for Author):

*This is an interesting study demonstrating a new pharmacological target to counteract the long-term negative cognitive effects of cannabis use in adolescence. Authors previously showed that 5-HT₆ receptors physically interact with mTOR to increase mTOR signalling and that 5HT₆ receptor antagonists could reverse cognitive and social interaction defects in schizophrenia mouse models (Meffre *et al.* 2012). In the present study they carry this analysis further focusing on another neurodevelopmental disorder: cannabis use during adolescence also known to increase mTOR signalling. THC administration from P30 to P45 increased mTOR-phosphorylation in the PFC and caused cognitive alterations at P60 (novel object recognition test and social interaction test). These effects were annulled by 5-HT₆ antagonist co-administered with THC, but not when the 5HT₆ antagonist were administered 15 days later. Additionally, authors provide some*

mechanistic insights on the cellular underpinnings of these long-term changes induced by THC. Using electrophysiological recordings of PFC slices, they show that adolescent THC reduces GABA input, and enhances excitatory drive onto layer V neurons. These neurons also show increase in their intrinsic excitability, that are reversed by a HCN1 channel antagonist. These effects are also dependent on the 5HT6R and appear to be downstream of mTOR signalling.

The paper is clearly written, well-illustrated with adequate controls and contributes to better understand long-term effects of cannabis in adolescents and proposes potentially important a new pharmacological target to reverse these effects.

My criticisms are minor and essentially suggestions to clarify some points.

We thank the reviewer for his interest in our study and positive comments.

1) Authors state that HT6 antagonists have no effect to reverse behaviour and mTOR in later administration protocols. However, the time frames of the experiments are not the same. Authors state that mTOR activation is persistent throughout life, however comparison of figures 1 and 2, shows that mTOR phosphorylation could decrease from P60 to P90, suggesting that it gradually wear off. Please comment.

In the late SB258585/rapamycin administration protocol (from P60 to P75), the biochemical and behavioral experiments were performed at P90, i.e. in the same time frame (15 days) after the last SB258585/rapamycin injection as in the adolescent administration protocol (last injection at P45, experiments performed at P60).

We have compared the stimulation of mTOR activity induced by THC delivered during adolescence and in adulthood and found not significant difference in mTOR stimulation between the two protocols: 2.61 ± 0.65 , $n=5$, fold increase in mTOR phosphorylation induced by adolescent THC administration vs. 1.99 ± 0.18 , $n=6$, fold increase induced by THC administration in adulthood, $p = 0.34$, unpaired Student's t-test.

2) On the same line: have the authors tried to administer THC from P60 to P75 to determine whether effects are really restricted to the juvenile period.

As requested by the reviewer, we administered THC between P60 and 75 and showed that this treatment does not induce a delayed (15 days after the last injection) mTOR activation and deficit in the novel object recognition task. This further supports the vulnerability of the juvenile brain to THC. These new data are illustrated on Supplementary Figure S6 and described in "Results", page 9.

3) Ephys data, suggest a reduced GABA input on layer 5 PFC neurons, and enhanced HCN1 channel expression. It would be nice if the authors could backup this interesting lead with morphology, or gene expression studies. Particularly since these developmental mechanisms are discussed

We now mention studies indicating that THC exposure during adolescence results in the premature pruning of spines and protracted atrophy of distal apical trees associated with impairment of LTD and alteration of synaptic markers (Rubino *et al.* Neurobiol. Dis., 73:60-9, 2015; Miller *et al.*, Mol. Psy. 24:588-600, 2019). These findings suggest that adolescent THC exposure reduces the complexity of pyramidal neurons, which might prematurely attenuate the capacity for plasticity in neural circuits central for normal adult behavior (see Discussion, page 15).

We also cite recent RNA-seq data obtained by Yasmin Hurd's group in rats treated with THC during adolescence (Miller *et al.* Molecular Psychiatry 24:588-600, 2019) which show a decrease in Raptor and HCN1 mRNA levels in THC-treated rats, compared with vehicle-treated rats, two weeks after the last THC administration. These findings (discussed on page 15-16) indicate that the overactivation of mTOR and the resulting dysregulation of HCN1 channel in THC-treated animals are not caused by an increase in their gene expression but rather from an increase in their catalytic (mTOR) or channel (HCN1) activity.

4) Rapid reversal of hyperexcitability was obtained with the HCN1 channel blocker ZD7288 on PFC slice. Can this compound be administered in vivo to test whether this might reverse

behavioural abnormalities; This seemed like an obvious experiment, but maybe not possible.

As anticipated by the reviewer, the HCN1 channel inhibitor ZD7288 could not be administered in vivo due to the deleterious influence on the cardiac function (Luo *et al.* Neuroscience 144:1477-85, 2007).

5) Discussion on the possible developmental impact of enhanced mTOR signaling under the control of 5-H6R seemed overly long and speculative since the authors do not provide new insight on these mechanisms.

As requested by the reviewer, we have reduced this part of the discussion.

Referee #3 (Remarks for Author):

In this study, Berthoux and colleagues investigate the long-term effects of chronic THC use in adolescent mice. In particular, they show that THC daily injections from Postnatal Day (P) 30 to P54 lead to

- increased mTOR activation in prefrontal cortex (but not hippocampus)*
- impaired behavior in the novel recognition task (THC-treated mice failed to recognize the novel object) and in the social preference task (THC - treated mice spend less time with the mouse compared to vehicle treated mice)*
- altered mIPSC (reduced) and mEPSC (increased) frequency in L5 pyramidal neurons*
- increased sag (due to HCN1 channel activity), leading to increased resting membrane potential and action potential threshold in L5 pyramidal neurons.*

Remarkably, simultaneous treatment during adolescence with rapamycin (mTOR inhibitor) or SB285585 (5HT6 inhibitor) rescues the biochemical, electrophysiological and behavioral phenotypes. Conversely, two weeks treatment with SB or RAPA from P60-75 does not normalize mTOR hyperactivation or deficits in novel object recognition.

This is a very timely, very well written and well-executed study, which sheds light on an important issue, namely the potential molecular mechanisms underlying the long-term effect chronic THC use in adolescence. I have relatively minor comments and suggestion that would help readers to evaluate and reproduce the data.

We thank the reviewer for his interest in our study and positive comments.

1) In methods section, the authors describe the two different vehicle solution used for THC and SB/RAPA. Are data regarding mice treated with THC and with the vehicle for RAPA/SB shown? Are data regarding mice treated with the vehicle for RAPA/SB alone? I do not think so. Please, correct methods accordingly by clarifying that the only control used are mice treated with the vehicle for THC alone or vehicle for THC+SB or vehicle for THC+RAPA.

All animals received the same number of injections. Correspondingly, control animals were successively injected with the vehicle used for THC and the vehicle used for SB258585/Rapamycin and are referred as "Veh/Veh" on the figures. Likewise, mice treated with the vehicle used for THC and SB/Rapa were referred to Veh/SB and Veh/Rapa, respectively, while mice treated with THC and the vehicle used for SB/Rapa were referred to THC/Veh and THC/Veh. This point has also been clarified in the "Materials and Methods" section (see page 20, paragraph Drugs & treatments).

2) The authors recorded 5 neurons from prelimbic or anterior cingulate cortex. Why were two regions chosen? Were n of recorded cells equally taken from these two regions? Are electrophys properties comparable? Please, add this information in the methods.

For each recording, we measure the coordinates of the recorded neurons using a micrometer (Hamamatsu). We verified these coordinates and we confirmed that pyramidal neurons were recorded from the prelimbic cortex only, in contrast to what we mentioned in the previous version of the manuscript. We apologize for this mistake.

This region of the rodent ventromedial prefrontal cortex is known to mediate cognitive functions, decision-making, and emotional regulation (Corbit et al. 2003, Behav Brain Res 146:145–157), making them a central component of mesolimbic and cortical circuitries whose disruption is implicated in the etiology of multiple psychiatric illnesses in humans.

3) In the methods/novel object recognition task, the authors write that 'Mice were extensively handled and habituated to the test'. Please, be more precise so other operators can reproduce the experiments. Authors also write that novel object recognition task was performed on day 3 and 4. What was the difference in the tests performed on day 3 and 4? Were completely different objects used? Where discrimination index combined on day 3 and 4?

More information was added in the “Materials and Methods” section (see page 22, paragraph Behavioral analysis). The habituation period was performed on days 1 and 2, the test on day 3, in contrast to what we mentioned in the previous version of the manuscript (day 4). We apologize for this mistake.

4) In the methods/social preference task. Were mice habituated to the chamber? If yes, for how long? How were congener mice chosen (what was the age range, for example).

Each tested mouse was placed for 10 min in the middle compartment of the three-chamber device, without the wall between the compartments to allow free access to the three compartments, before the tests. Congeners were same age and same sex as the tested mice. The information was added to the “Materials and Methods” section (see pages 22-23, paragraph Social preference task).

5) Were the same mice tested for novel object recognition and social preference? If yes, how long did operators waited between the two tests?

We indeed used the same cohorts for both tests. One of them was performed on week 1 and the second on week 2. The objects used for the novel object recognition and the sociability tests were different.

6) Where mice of both sexes tested in behaviour and in all other experiments? Add this important information.

We used mice of both sexes for all experiments. This information was already mentioned in the “Animals” sub-chapter of the “Materials and Methods” section, but we added a sentence to clarify this point (see page 19, paragraph Animals).

We did not notice any significant difference in the results between both sexes.

7) In the methods section, please add all catalog numbers of antibodies and other critical reagents.

As requested by the reviewer, the information was added in the “Materials and Methods” section.

8) Results section. As a general comment, please add exact N of mice or n of recorded cell/ N of mice for each experimental group. It would be helpful to plot all data set as it was done for the behavioural data, namely showing all data point and not only average values.

As requested, we plotted all data sets on the figures for the electrophysiological experiments and added the N of mice and n of recorded cells when they were missing.

9) Figure 3. N of recorded neurons is missing. Please see comment #6

The information was added in the legend.

10) Figure 4. Please add exact n from exact N of mice, per groups.

The information was added in the legend.

11) Fig 2 and 3. It would be helpful to plot raw values for all data groups and not only values normalized on vehicle.

The normalized plots have been replaced by plots with raw data.

12) Supplemental Figures S1, S2, S3 and S4. Please write that mice were treated with Veh alone or Veh+SB or Veh+RAPA. Figure S3, n of recorded cells in each experimental group is missing. Figure S4, n of cells and N of mice is missing. S5, n of recorded cells is missing. Please see and apply comment #6 to all supplemental figures, too.

The missing information were added in the legends of Supplementary Figures.

13) Is locomotion and anxiety behavior similar in vehicle and THC-treated mice? These parameters can be evaluated using open field and elevated plus maze tests and are critical to interpret altered social behavior.

We have evaluated these parameters using the elevated plus maze test (anxiety) and the cyclotron (locomotion). We did not find any significant difference between THC- or vehicle-injected mice. These new data are illustrated on Supplementary Figure S5 and described in "Results", page 7.

14) Data showing whether the activity of the HCN1 channels is rescued by RAPA or SB treatment during adolescence would further strengthen the authors' conclusions.

As requested by the reviewer, we recorded the voltage sag in layer V pyramidal neurons from THC-injected mice treated with SB258585 or rapamycin during adolescence. The sag amplitude was similar to the one recorded in neurons from vehicle-injected mice confirming that the early blockade of the 5-HT6/mTOR signaling pathway prevents the neuronal alterations underlying HCN1 dysregulation induced by chronic THC administration during adolescence. These new data are illustrated on Figure 4E and described in "Results", page 11.

15) Figure 2. It would be interesting, if possible, to add data for social discrimination too, for consistency.

As requested by the reviewer, we performed the social discrimination test. Mice exposed to THC during adolescence showed an altered performance in social discrimination, compared to vehicle-injected mice (discrimination index: -0.04 ± 0.09 and 0.32 ± 0.06 for THC+Vehicle and for Vehicle+Vehicle conditions, respectively, $p < 0.01$). This deficit was prevented by injecting SB258585 or rapamycin during adolescence (discrimination index: 0.28 ± 0.06 and 0.25 ± 0.07 for THC+SB and for THC+rapa conditions, respectively, $p > 0.05$ vs. Vehicle mice). These new data are illustrated on Figure 1 and described in "Results", pages 7-8.

2nd Editorial Decision

21st Nov 2019

Thank you for the submission of your manuscript to EMBO Molecular Medicine. We have now received the two reports from the two referees who were asked to re-evaluate your revision.

You will see that ref. 3 is now fully supportive. Unfortunately, ref. 1 feels like although most of the concerns were addressed, a few remains, and s/he would like to see these points of criticisms answered. The comments are very much self-explanatory, and I hope that you will also see them as constructive, meant to strengthen the conclusions.

As you may know, we normally do not allow a second round of revision. This said, we would like to give you a final chance to address the referee's critique satisfactorily. Please proceed with revision and resubmit your revised article following the same steps as you did before. As for timeline, please resubmit as soon as you will have addressed the comments listed below.

***** Reviewer's comments *****

Referee #1 (Remarks for Author):

The Authors have performed a significant number of experiments to improve upon many aspects of the manuscript. This reviewer quite clearly appreciates the amount of work put into addressing specific queries. Therefore, the questions listed here solely relate to those originally asked and for which answers do not seem to be adequate.

1. The Authors press on that their findings are "proof of concept" and admittedly the mechanism "remains partly characterised". In reading the manuscript, one wonders why a summary figure that would show the cellular players, intracellular events and drug action sites (probably also using question marks for (and if) unknown steps) is missing. A comprehensive concluding figure would certainly help the general reader.

2. One point of confusion I fear comes from the histochemical data. The Authors state that 5HT6 receptors are postsynaptic. Looking at the single representative image I find absolutely no proof of that. In fact, the staining could equally be presynaptic enwrapping a somatodendritic domain of a cell (SIFig 4), even if I agree with that the bulk of CB1R seem separately partitioned from 5HT6 receptors in that picture. Minimally, the authors should use a presynaptic marker like synaptobrevin, syntaxin and a postsynaptic one like PSD95 or even better a pyramidal cell marker (SMI311, SMI32 or alike) to substantiate the anatomical claim. Elucidating the anatomy at this level is a must and the figure, once ready, should be part of the main figure set. (another option could be to perform FISH using HCR/RNA-scope for 5HT6). Quantification of the amount of neurons receiving both 5HT6+ and/or CB1R+ inputs will be essential, also including interneurons (I am fine if a general interneuron marker is used), and their location.

3. Following on the above and my original question 4: "where does defunct GABA input originate from" is unresolved. At the very least, VGAT/CB1R and VGAT/5HT6 histochemistry would be amenable and help resolving network specifics. The CB1R+ punta seem like those of CCK+ interneurons, which could be a bias of the antibody used (please clean up the paper (blue text) on this description and conclusions, too). A paper from Beat Lutz earlier showed that only 20-30% of 5HT neurons express CB1Rs. Therefore, and if any of the 5HT6 receptors are presynaptic, then their colocalization with CB1Rs shall per se not be excluded.

4. My original question 6 focused on "sustained site-directed phosphorylation", which the Authors do not seem to address. I understand that 5HT6 receptors acutely signal through mTOR and that is absolutely fine. But what dictates the prolonged change.

Once resolving these points, I certainly be willing to recommend the paper to EMBO Mol Med.

Referee #3 (Remarks for Author):

The authors have responded to all my comments and suggestions.

2nd Revision - authors' response

9th Jan 2020

Response to the reviewer's comments

The Authors have performed a significant number of experiments to improve upon many aspects of the manuscript. This reviewer quite clearly appreciates the amount of work put into addressing specific queries. Therefore, the questions listed here solely relate to those originally asked and for which answers do not seem to be adequate.

1. *The Authors press on that their findings are "proof of concept" and admittedly the mechanism "remains partly characterised". In reading the manuscript, one wonders why a summary figure that would show the cellular players, intracellular events and drug action sites (probably also using question marks for (and if) unknown steps) is missing. A comprehensive concluding figure would certainly help the general reader.*

As requested by the Reviewer, we provide a model illustrating the putative mechanisms elicited by chronic THC consumption during adolescence, that lead to a long-lasting, 5-HT₆ receptor-dependent mTOR activation in the prefrontal cortex and cognitive deficits in adulthood (see Figure 7).

2. *One point of confusion I fear comes from the histochemical data. The Authors state that 5HT6 receptors are postsynaptic. Looking at the single representative image I find absolutely no proof of that. In fact, the staining could equally be presynaptic enwrapping a somatodendritic domain of a cell (SIFig 4), even if I agree with that the bulk of CB1R seem separately partitioned from 5HT6 receptors in that picture. Minimally, the authors should use a presynaptic marker like synaptobrevin, syntaxin and a postsynaptic one like PSD95 or even better a pyramidal cell marker (SMI311, SMI32 or alike) to substantiate the anatomical claim. Elucidating the anatomy at this level is a must and the figure, once ready, should be part of the main figure set. (another option could be to perform FISH using HCR/RNA-scope for 5HT6). Quantification of the amount of neurons receiving both 5HT6+ and/or CB1R+ inputs will be essential, also including interneurons (I am fine if a general interneuron marker is used), and their location.*

We performed immunohistochemistry experiments on prefrontal cortex slices to show presynaptic (assessed by Bassoon immunostaining) vs. postsynaptic (assessed by PSD-95 immunostaining) localization of CB₁ and 5-HT₆ receptors, respectively. The data (illustrated on a new Figure 2) indicate that 5-HT₆ receptors are strongly co-localized with PSD-95, whereas CB₁ receptors are mostly co-localized with the presynaptic protein Bassoon (see also Supplementary Figure 4 for the quantification of protein co-localization and "Results", page 7 (first paragraph), even though a small fraction of CB₁ receptors is also found at the post-synapse, consistent with previous findings (Maroso et al., Neuron, 2016; 89:1059-1073).

3. *Following on the above and my original question 4: "where does defunct GABA input originate from" is unresolved. At the very least, VGAT/CB1R and VGAT/5HT6 histochemistry would be amenable and help resolving network specifics. The CB1R+ punta seem like those of CCK+ interneurons, which could be a bias of the antibody used (please clean up the paper (blue text) on this description and conclusions, too). A paper from Beat Lutz earlier showed that only 20-30% of 5HT neurons express CB1Rs. Therefore, and if any of the 5HT6 receptors are presynaptic, then their colocalization with CB1Rs shall per se not be excluded.*

We provide immunohistochemistry experiments on prefrontal cortex slices (also illustrated on Figure 2), which show a strong co-localization of CB₁ receptor and GAD65 immunostaining, indicating that the majority of CB₁ receptors is located on GABAergic terminals, while only a very small fraction of 5-HT₆ receptors is co-localized with GAD65.

Co-immunostaining of CB₁ receptor and the serotonin transporter (SERT) also showed the presence of a small fraction of receptors in 5-HT terminals (see Figure 2). Due to specie incompatibility of the SERT and 5-HT₆ receptor antibodies available, we could not perform co-staining of 5-HT fibers and the receptor. However, a previous *in situ* hybridization study showed that serotonergic neurons do not express 5-HT₆ receptor mRNA, suggesting that the 5-HT₆ receptor is not a presynaptic autoreceptor on serotonergic neurons (Helboe et al., Neuroscience 310: 442-454, 2015).

Collectively, these anatomical studies indicate that CB₁ and 5-HT₆ receptors exhibit distinct neuronal localization in the prefrontal cortex.

Finally, to ensure the absence of bias in CB₁ receptor immunostaining pattern due to the antibody used, we compared CB₁ receptor immunostaining obtained with the guinea pig polyclonal antibody (GP) used in the manuscript and a rabbit polyclonal antibody (Rb, both obtained from Frontier Institute co., Ltd). As shown in the enclosed figure, a strong co-localization of both immunostainings was observed (Mander's coefficient: 0.960 ± 0.009).

4. My original question 6 focused on "sustained site-directed phosphorylation", which the Authors do not seem to address. I understand that 5HT₆ receptors acutely signal through mTOR and that is absolutely fine. But what dictates the prolonged change.

We agree with the reviewer that a full characterization of the mechanism underlying the long-lasting activation mTOR in THC-injected mice is an important issue. Here, we provide a first evidence that it results from a sustained activation of 5-HT₆ receptors by endogenously released 5-HT, as it was prevented by the administration of a 5-HT₆ receptor neutral antagonist during adolescence. The enhanced release of 5-HT in the prefrontal cortex of adolescent THC-injected animals likely results from CB₁ receptor-mediated decrease in GABA release and the disinhibition of 5-HT terminals. The mechanism underlying the persistent activation of mTOR in adulthood remains more uncertain. We hypothesize that it results from a remodeling of the cortical network that might affect the serotonergic system itself, as a consequence of the non-physiological activation of mTOR at a critical period of PFC maturation. We have modified the discussion to clarify that point (see pages 14 and 15).

3rd Editorial Decision

12th Feb 2020

Thank you for the submission of your revised manuscript to EMBO Molecular Medicine and for your patience while we were completing the review process. We have now received the enclosed reports from the three referees that were asked to re-assess it. As you will see the reviewers are now globally supportive and I am pleased to inform you that we will be able to accept your manuscript pending the following final amendments:

1) Please address the comments made by the referees in writing. We would like you to provide quantification of immunological experiments as suggested by referee #3 and rework the figure model as suggested by referees #2 and #3 to include "?" or any other way to make it clear that some aspects of the model are only speculative and not demonstrated here (see the issues highlighted by referee #1).

***** Reviewer's comments *****

Referee #1 (Remarks for Author):

The authors have made an attempt to address my queries. Unfortunately, I feel the anatomy work is short of depth, neither temporal resolution nor quantification. The other replies are also ambiguous. There is a number of central hypotheses that, with the eye of a developmental neurobiologist, are absolutely not justified. Some of the major assertions are:

- 1: ...the latter observation indicates it might result from a non-physiological 5-HT₆ receptor activation by endogenously released 5-HT rather than constitutive activity, which might be caused by CB₁ receptor-mediated decrease in GABA release and the disinhibition of 5-HT terminals (Figure 7) in the prefrontal cortex,
- 2: ... The mechanism underlying the persistent activation of mTOR in adulthood remains uncertain,

3:the resulting non-physiological mTOR activation interferes with the maturation of the GABAergic system.

For #1, there is no direct evidence of enhanced 5-HT release. For #2, uncertainty is clearly disallowed at this level, for #3, one would want to see that those CB1R-positive GABA terminals are indeed rearranged (less, more, targeting deficit, power of modulation?). The authors cite a breadth of literature on how THC affects PFC morphology yet there is no evidence of dendritic vs. synaptic reorganisation at any level.

Lastly, I fear the summary figure is conceptually misleading. It places GABAARs onto glutamatergic and 5-HT PRESYNAPSES for which I cannot recall any evidence, nor can I justify (from the present data) the power of such signalling loop in adolescence. The suggestion then that such SYNAPSE level integration (including spillover? cross-modulation?) would then be permanently effective is to me quite clearly unrealistic to define long-term PFC changes upon THC treatment. My worry is that this figure is because the authors accumulated mismatching data that just simply do not fill the gaps appropriately.

In sum, I think the paper suffers from wanting to dissect a developmental (and therefore temporally precise) mechanism without using several time-points as developmental read-outs and even with more and more data it stretches the boundaries of wishful imagination rather than rigorous reasoning. For being "molecular" in medicine I therefore find little if any justification.

Referee #2 (Remarks for Author):

I went back briefly to the paper although I could not recall all the details and did not look back at all reviewer's comments.

For me the very strong point of the paper is the novel demonstration of a requirement of 5-HT₆ for the specific developmental effects of cannabis exposure during adolescence. This is convincingly demonstrated with both pharmacology and genetics and concern the long-term effects at 3 different levels : 1) cognitive & Social behaviour; 2) biochemical changes that are relevant to developmental changes : activation of the mTOR pathway ; 3) Cell physiology with changes of neuronal excitability of PFC neurons, with changes in intrinsic properties of the pyramidal neurons and Reduction of LTD on pyramidal neurons. Importantly They show that these effects are not observed when the same pharmacological agents are administered in adulthood.

For me this is new evidence, which is very rigorously obtained. It is interesting from a translational point of view to help understand some of the long-lasting changes of cannabis in adolescence, and for fundamental research to show how pharmacological interactions change dramatically during development, even at late developmental stages such as adolescence.

I would agree with reviewer 1 that the mechanistic explanation has still many holes at this stage and unclear for instance that cannabis modulates 5-HT release, However authors do make a case that 5-HT₆ is mainly postsynaptic on PFC neurons, while Cb1 is presynaptic and essentially present on GABA presynaptic terminals. Thus the 2 receptors are not on the same neurons and likely involve different cell types, and thus there is the beginning of a model. The model should include some question marks, as there is always a great level of uncertainty in models (you would be surprised going back to many eminent publications). These models are just there to help us to integrate the different experiments.

In conclusion I am very supportive of publication, with possibly some toning down of mechanistic interpretations that should be presented as working hypotheses for the future rather than demonstrated facts.

Referee #3 (Remarks for Author):

I re-read carefully all 3 version of the manuscripts and all answer to reviewers.

In summary, I strongly believe that this paper show sufficiently novelty to justify publication in EMBO Mol Medicine (as stated as well by referee #1 when he reviewed the Version2 of this manuscript). I also believe that it is always possible to ask for more and more experiments, since there is so much that we do not know about the neurobiology mechanisms in questions, however I think the authors did enough to answer the referee's concern.

here my point by point answer to referee #1 comments:

about temporal resolution or quantification of the data

I agree that temporal resolution would be interesting to address in future studies, but it is beyond the scope of this study right now. Please note that temporal resolution or using several time-points as developmental read-outs was not asked in the two previous reviews rounds, so it should **not *be* asked now at the third round, in my opinion.

I would suggest that the authors provide quantification for immuno experiments in figure 2 (new data provided by the authors to answer to referee #1's second review round). Personally, I like the idea to see data quantification wherever possible, because it strengthens the results.

about an incompletely resolved mechanism

Please note that [...] the authors are already very cautious in their statements. (For example, they use the verb "might" and state clearly the mechanism underlying mTOR persistent activation remain uncertain). I do not find that the authors oversold their speculations.

about the model figure

I would ask the authors to rework the figure by referring to the papers/data from where they got the different info and clearly stating what is just speculative at this point.

Finally, since the authors performed all the experiments asked [...] in round two of the review process (see point 2 and 3 of second revision by Referee#1) and the different time points were not asked at that time point of the revision process, I would not consider it fair to add this new issue right now.

3rd Revision - authors' response

5th Mar 2020

Points raised by Referee #1:

The authors have made an attempt to address my queries. Unfortunately, I feel the anatomy work is short of depth, neither temporal resolution nor quantification.

We have done the requested experiments regarding the synaptic localization of both receptors and quantified their co-localization ratio with synaptic markers (the quantification was illustrated on a supplementary figure in the previously revised version (Figure EV2) and has now been moved to Figure 2 along with the immunofluorescence images).

The temporal resolution was not asked by Referee #1 in his previous review and, as outlined by the two other referees, is beyond the scope of the present study.

Overall, we were a bit surprised and disappointed to read that the referee was not convinced by these new anatomical data that entirely addressed what he/she asked in his/her second review, and that he/she changed his/her opinion regarding the publication of our work. One explanation might be that his/her hypothesis was a co-expression of CB₁ and 5-HT₆ receptors on the same neurons and the existence of a crosstalk mechanism between both receptor subtypes, whereas the anatomical data added during the last revision clearly show that both receptors are located of different neuronal populations.

The other replies are also ambiguous. There is a number of central hypotheses that, with the eye of a developmental neurobiologist, are absolutely not justified. Some of the major assertions are:

1: ...the latter observation indicates it might result from a non-physiological 5-HT₆ receptor activation by endogenously released 5-HT rather than constitutive activity, which might be caused by CB₁ receptor-mediated decrease in GABA release and the disinhibition of 5-HT terminals (Figure 7) in the prefrontal cortex,

2: ... The mechanism underlying the persistent activation of mTOR in adulthood remains uncertain,

3: ...the resulting non-physiological mTOR activation interferes with the maturation of the GABAergic system.

For #1, there is no direct evidence of enhanced 5-HT release.

We agree with the referee that there is no direct evidence of enhanced 5-HT release. However, our results obtained with CPPQ, a neutral 5-HT₆ receptor antagonist that specifically inhibits agonist-induced receptor activation and not its constitutive activity, strongly support that the non-physiological receptor activation in mice treated with THC during adolescence does result from an increase in 5-HT release.

For #2, uncertainty is clearly disallowed at this level,

The cited sentence is part of the discussion chapter. Our study clearly demonstrates the role of 5-HT₆ receptors in the mTOR activation during adolescence and the contribution of this pathway in the late-onset cognitive deficits in THC-treated mice. We agree with the referee that understanding the mechanisms underlying the persistent mTOR activation in adulthood is also an important question, but it is highly challenging and likely requires years of work to be entirely solved. However, we feel useful to propose potential mechanisms in the discussion, even though we acknowledge that some aspects remain uncertain.

for #3, one would want to see that those CB₁R-positive GABA terminals are indeed rearranged (less, more, targeting deficit, power of modulation?). The authors cite a breadth of literature on how THC affects PFC morphology yet there is no evidence of dendritic vs. synaptic reorganization at any level.

In his previous review, Referee 1 only asked us to use postsynaptic and presynaptic markers to characterize the synaptic localization of 5-HT₆ receptors and to determine whether CB₁ and 5-HT₆ receptors might be co-localized. We have done the requested experiments and showed that 5-HT₆ receptors are mostly located post-synaptically on pyramidal neurons, while CB₁ receptors are mainly presynaptic on GABAergic neurons (Figure 2). We agree that investigating the rearrangement of CB₁-positive GABA terminals in a model of chronic consumption of THC during adolescence would be an interesting question, but again this is beyond the scope of the current study. As stated by the other referees, it is always possible to ask for more and more experiments, since there is so much that we do not know about the neurobiological changes induced by THC at this period of PFC maturation.

Lastly, I fear the summary figure is conceptually misleading. It places GABAARs onto glutamatergic and 5-HT PRESYNAPSES for which I cannot recall any evidence, nor can I justify (from the present data) the power of such signaling loop in adolescence. The suggestion then that such SYNAPSE level integration (including spillover? cross-modulation?) would then be permanently effective is to me quite clearly unrealistic to define long-term PFC changes upon THC treatment. My worry is that this figure is because the authors accumulated mismatching data that just simply do not fill the gaps appropriately.

The model figure was added to address a query of the referee asking for “a summary figure that would show the cellular players, intracellular events and drug action sites with marks for unknown steps”. This model established from the one proposed by Puighermanal *et al.* (Nature Neurosci. 12(9):1152-8, 2009), is an attempt of mechanistic explanation of the initial THC effects occurring during adolescence that underlie the long-term changes in the PFC of THC-treated mice. This is only a model helping to integrate the different

experiments with uncertain aspects outlined by question marks, such as the increased release of 5-HT in THC-treated mice and the presence of GABA-A receptors on 5-HT terminals (see Figure 7). GABA-A receptors were placed onto glutamatergic fibers in line with previous studies demonstrating their expression on glutamatergic terminals (Yamamoto *et al.*, Brain Res Bull., 84(1):22-30, 2011; Ruiz *et al.*, Nature Neurosci., 13(4):431-8, 2010; Alle & Geiger, J. Neurosci., 27(4):942-50, 2007). Likewise, the presence of GABA-A receptors onto serotonergic terminals was previously suggested by the study of Cerrito *et al.* (J. Neurosci Res., 51(1):15-22, 1998). These references are now quoted the figure legend.

Points raised by Referee #2:

I went back briefly to the paper although I could not recall all the details and did not look back at all reviewer's comments.

For me the very strong point of the paper is the novel demonstration of a requirement of 5-HT₆R for the specific developmental effects of cannabis exposure during adolescence. This is convincingly demonstrated with both pharmacology and genetics and concern the long-term effects at 3 different levels: 1) cognitive & Social behavior; 2) biochemical changes that are relevant to developmental changes : activation of the mTOR pathway ; 3) Cell physiology with changes of neuronal excitability of PFC neurons, with changes in intrinsic properties of the pyramidal neurons and Reduction of LTD on pyramidal neurons. Importantly They show that these effects are not observed when the same pharmacological agents are administered in adulthood.

For me this is new evidence, which is very rigorously obtained. It is interesting from a translational point of view to help understand some of the long-lasting changes of cannabis in adolescence, and for fundamental research to show how pharmacological interactions change dramatically during development, even at late developmental stages such as adolescence.

I would agree with reviewer 1 that the mechanistic explanation has still many holes at this stage and unclear for instance that cannabis modulates 5-HT release. However, authors do make a case that 5-HT₆R is mainly postsynaptic on PFC neurons, while Cb1 is presynaptic and essentially present on GABA presynaptic terminals. Thus the 2 receptors are not on the same neurons and likely involve different cell types, and thus there is the beginning of a model.

The model should include some question marks, as there is always a great level of uncertainty in models (you would be surprised going back to many eminent publications). These models are just there to help us to integrate the different experiments.

We thank the referee for his/her positive and constructive comments during the reviewing/revision process and his/her support for the publication of the manuscript.

As suggested by the referee, we added some question marks for the more uncertain aspects of the model, such as the increased release of 5-HT in THC-treated mice and the presence of GABA-A receptors on 5-HT terminals (see Figure 7).

Points raised by Referee #3

I re-read carefully all 3 version of the manuscripts and all answer to reviewers.

In summary, I strongly believe that this paper show sufficiently novelty to justify publication in EMBO Mol Medicine (as stated as well by referee #1 when he reviewed the Version2 of this manuscript). I also believe that it is always possible to ask for more and more experiments, since there is so much that we do not know about the neurobiology

mechanisms in questions, however I think the authors did enough to answer the referee's concern.

We thank the referee for his/her positive and constructive comments during the reviewing/revision process and his/her support for the publication of the manuscript.

here my point by point answer to referee #1 comments:

about temporal resolution or quantification of the data

*I agree that temporal resolution would be interesting to address in future studies, but it is beyond the scope of this study right now. Please note that temporal resolution or using several time-points as developmental read-outs was not asked in the two previous reviews rounds, so it should *not* be asked now at the third round, in my opinion.*

I would suggest that the authors provide quantification for immuno experiments in figure 2 (new data provided by the authors to answer to referee #1's second review round). Personally, I like the idea to see data quantification wherever possible, because it strengthens the results.

Quantification of immunological experiments was already provided on a supplementary figure of the previous version. To facilitate the reading of the manuscript, we now included it along with the immunofluorescence images on Figure 2, as suggested by the referee.

about an incompletely resolved mechanism

Please note that [...] the authors are already very cautious in their statements. (For example, they use the verb "might" and state clearly the mechanism underlying mTOR persistent activation remain uncertain). I do not find that the authors oversold their speculations.

about the model figure

I would ask the authors to rework the figure by referring to the papers/data from where they got the different info and clearly stating what is just speculative at this point.

We reworked the model figure and added some question marks for the more uncertain aspects of the model, such as the increased release of 5-HT in THC-treated mice and the presence of GABA-A receptors on 5-HT terminals (see Figure 7). We also quoted previously published papers from where information was taken to build the model in the figure legend.

Finally, since the authors performed all the experiments asked [...] in round two of the review process (see point 2 and 3 of second revision by Referee#1) and the different time points were not asked at that time point of the revision process, I would not consider it fair to add this new issue right now.

Corresponding Author Name: Carine BECAMEL

Manuscript Number: EMM-2019-10605-V2